# The Immunomodulatory Effect of IrSPI, a Tick Salivary Gland Serine Protease Inhibitor Involved in *Ixodes ricinus* Tick Feeding

**DOI:** 10.3390/vaccines7040148

**Published:** 2019-10-12

**Authors:** Adrien A. Blisnick, Ladislav Šimo, Catherine Grillon, Fabienne Fasani, Sébastien Brûlé, Bernard Le Bonniec, Eric Prina, Maud Marsot, Anthony Relmy, Sandra Blaise-Boisseau, Jennifer Richardson, Sarah I. Bonnet

**Affiliations:** 1UMR BIPAR, Animal Health Laboratory, INRA, ANSES, Ecole Nationale Vétérinaire d’Alfort, Université Paris-Est, 94701 Maisons-Alfort CEDEX, France; ablisnick@gmail.com (A.A.B.); ladislav.simo@vet-alfort.fr (L.Š.); 2Centre de Biophysique Moléculaire—UPR 4301 CNRS, 45000 Orléans, France; catherine.grillon@cnrs-orleans.fr (C.G.); fabienne.fasani@cnrs-orleans.fr (F.F.); 3Plateforme de Biophysique moléculaire, Institut Pasteur, UMR 3528 CNRS, 75015 Paris, France; sebastien.brule@pasteur.fr; 4INSERM UMR-S1140, Faculté de Pharmacie Université Paris Descartes, Sorbonne Paris Cité, 75270 Paris CEDEX 06, France; bernard.le-bonniec@parisdescartes.fr; 5Unité de Parasitologie moléculaire et Signalisation—INSERM U1201, Institut Pasteur, 75015 Paris, France; eric.prina@pasteur.fr; 6Unité EPI, Animal Health Laboratory, INRA, ANSES, Ecole Nationale Vétérinaire d’Alfort, Université Paris-Est, 94701 Maisons-Alfort CEDEX, France; maud.marsot@anses.fr; 7UMR Virologie 1161, Animal Health Laboratory, INRA, ANSES, Ecole Nationale Vétérinaire d’Alfort, Université Paris-Est, 94701 Maisons-Alfort CEDEX, France; Anthony.RELMY@anses.fr (A.R.); Sandra.BLAISE-BOISSEAU@anses.fr (S.B.-B.); jennifer.richardson@vet-alfort.fr (J.R.)

**Keywords:** *Ixodes ricinus*, tick–host–pathogen interactions, anti-tick vaccine, serine protease inhibitor, immunomodulator, macrophages, lymphocytes

## Abstract

Ticks are the most important vectors of pathogens affecting both domestic and wild animals worldwide. Hard tick feeding is a slow process—taking up to several days—and necessitates extended control over the host response. The success of the feeding process depends upon injection of tick saliva, which not only controls host hemostasis and wound healing, but also subverts the host immune response to avoid tick rejection that creates a favorable niche for the survival and propagation of diverse tick-borne pathogens. Here, we report on the molecular and biochemical features and functions of an *Ixodes ricinus* serine protease inhibitor (IrSPI). We characterize IrSPI as a Kunitz elastase inhibitor that is overexpressed in several tick organs—especially salivary glands—during blood-feeding. We also demonstrated that when IrSPI is injected into the host through saliva, it had no impact on tissue factor pathway-induced coagulation, fibrinolysis, endothelial cell angiogenesis or apoptosis, but the protein exhibits immunomodulatory activity. In particular, IrSPI represses proliferation of CD4^+^ T lymphocytes and proinflammatory cytokine secretion from both splenocytes and macrophages. Our study contributes valuable knowledge to tick-host interactions and provides insights that could be further exploited to design anti-tick vaccines targeting this immunomodulator implicated in *I. ricinus* feeding.

## 1. Introduction

Ticks are obligate hematophagous arthropods able to feed on diverse vertebrate hosts including mammals, birds, amphibians, and reptiles. They cause substantial economic losses in the livestock industry due to blood spoiling and secondary infection of bite wounds, which consequently decrease food production and the value of leather. Crucially, ticks are excellent vectors for pathogens, including bacteria, parasites, and viruses. They are—after mosquitoes—the second-most common vector of human pathogens and the first for animal health [1]. In particular, the hard tick *Ixodes ricinus* is the most widespread and abundant tick species in Europe, as well as the most efficient transmitter of pathogens that significantly impact both human and animal health [2]. This tick species follows a three-host life cycle where each of the three distinct life stages (larvae, nymph and adult) feeds only once on its respective host.

The hard tick blood meal’ is one of the most extended processes of all blood-feeding arthropods. Several hard tick species feed over ten days as adults, thus, causing severe host tissue damage. The success of the blood meal is critically dependent on saliva, which represents the primary tick-host interface (reviewed in [3]). Indeed, tick saliva possesses several essential properties that facilitate feeding, including an ability to repress host pain at the tick bite site, thus, preventing early detection by the host. Besides, tick saliva can counteract host hemostatic and immunological responses at the site of injury. Ticks are pool-feeders, meaning that they create a hemorrhagic blood pool by alternating blood absorption and saliva injection. As such, the saliva’s anti-coagulant properties maintain the blood pool in a fluid state by blocking blood coagulation cascade activation and promoting fibrinolysis. Moreover, tick saliva impedes the host’s innate immune system, by preventing complement activation and immune cell recruitment, thus, precluding tissue remodeling and angiogenesis, as well as local inflammation. Tick saliva also impairs anti-tick adaptive immune responses, by repressing lymphocyte recruitment at the tick bite site. Finally, these tick bite-associated processes create an environment favoring the transmission of tick-borne pathogens (TBP) to the host.

A better understanding of tick-host-pathogen interactions is central to the development of improved tick and tick-borne disease (TBD) control methods [4]. In order to identify tick factors involved in both tick biology and pathogen transmission, we used next-generation sequencing (NGS) to compare gene expression in salivary gland (SG) tissue from uninfected and *Bartonella henselae*-infected *I. ricinus* ticks [5]. This gram-negative bacterium is responsible for cat scratch disease in humans and is transmitted from cat to cat by fleas. Although the majority of human cases are due to scratches and bites from infected cats, we have previously demonstrated that *I. ricinus* can also directly transmit this pathogen [6]. A small percentage of tick genes (5.6%) were found to be modulated by this bacterial infection. The most overexpressed gene—that we named *Ixodes ricinus* serine protease inhibitor (*IrSPI*)—encoded a putative serine protease inhibitor. Preliminary RNA interference (RNAi) functional analysis revealed that *IrSPI* is implicated in tick feeding, as well as in *B. henselae* infection of tick SG, rendering it a promising target as an anti-*I. ricinus* vaccine. It is also known that several serine protease inhibitors (SPI) are involved in the aforementioned tick saliva processes that maintain blood pools, further justifying our interest in IrSPI [7]. Serine protease inhibitors are one of the leading protein families injected into the host by ticks during blood-feeding, as inhibited proteins play key roles in regulating host coagulation, angiogenesis, hemostasis, and immune responses. Moreover, in ticks, several serine protease inhibitors are involved in blood digestion, the tick’s innate immune system, tick reproduction, as well as pathogen transmission [7].

In this study, we elucidate the IrSPI role both in tick biology and at the interface of the tick and the vertebrate host during feeding. In silico characterization of IrSPI confirmed that it belongs to the Kunitz type I family of serine protease inhibitors. We showed that *IrSPI* is induced during blood-feeding and that the protein is probably injected into the host, suggesting involvement in modulating host responses to tick feeding. We observe that the elastase is inhibited by recombinant IrSPI, whereas, neither the tissue factor pathway blood coagulation, the fibrinolysis, the endothelial cell apoptosis nor the angiogenesis is affected. Finally, we demonstrate that IrSPI modulates the host immune response by reducing the proliferative capacity of T lymphocytes and by altering the cytokine expression profile of both macrophages and splenocytes.

## 2. Material and Methods

### 2.1. Ethics

This study was carried out in strict accordance with good animal care practices recommended by the European guidelines. The local ethics committee approved animal experiments involving rabbits and mice used for tick feeding for animal experimentation (ComEth Anses/ENVA/UPEC; Permit Number 2015091413472401). Rabbits used for feeding pathogen-free ticks have been put up for adoption via the White Rabbit Association, Paris, France.

### 2.2. Ticks

This study used *I. ricinus* ticks either collected from the field, or from a pathogen-free colony. Questing ticks (nymphs and adults) were collected by flagging in the Sénart forest, located south-east of Paris (France), as previously described [8]. All pathogen-free *I. ricinus* ticks were derived from a laboratory colony reared at 22 °C in 95% relative humidity, with 12 h light/dark cycles as previously described [9].

### 2.3. Cloning and Sequencing of IrSPI cDNA

Following NGS of the *I. ricinus* salivary gland transcriptome, we had previously generated a total of 24,539 isotigs [5]. Of these, the *IrSPI* coding sequence (GenBank KF531922.2) without the stop codon (282 bp) was amplified with primers ID23822F (5′-ACATCTACCGTTCAAGATGAAG -3′) and ID23822R (5′-TCTAAGTGCCTTGCAGTAGTC-3′), and then cloned into the pET28 vector following the manufacturer’s instructions (Novagen, Merck KGaA, Darmstadt, Germany). After transformation of *Escherichia coli* TOP10 bacteria (ThermoFisher Scientific, Waltham, MA, USA) using plasmid and kanamycin selection, correct *IrSPI* sequence insertion was validated by PCR and sequencing.

### 2.4. Recombinant IrSPI Protein Production

The open reading frame (ORF) of IrSPI lacking the 5′ sequence encoding its signal peptide (MKATLVAICFFAAVSYSMG) was synthetized (Eurofins Scientific, Luxembourg) as a fusion protein with sequences encoding Twin-Strep-Tag (WSHPQFEKGGGSGGGSGGGSWSHPQFEK) and enterokinase (DDDDDK). The BglII and EcoRI restriction sites were used to insert the sequence into the pMT/BIP/V5-His plasmid (ThermoFisher, Waltham, MA, USA). The recombinant IrSPI protein was produced by the Recombinant Proteins in Eukaryotic Cells Platform, Pasteur Institute, Paris, France. Briefly, Drosophila S2 cells (Invitrogen, Carlsbad, CA, USA) adapted to serum-free Insect Xpress (Lonza, Basel, Switzerland) medium were co-transfected with pMT/BIP/IrSPI and the pCoPURO vector (Addgene, Cambridge, MA, USA) conferring resistance to puromycin, using the cellfectin II transfection reagent (Invitrogen, Carlsbad, CA, USA). The upstream pMT/BiP/V5-His *Drosophila* BiP secretion signal enables the recombinant protein to enter the S2 cell secretory pathway for recovery in the culture medium. IrSPI expression was driven by the metallothionein promoter after induction by 5 µM CdCl_2_. Clarified cell supernatants were concentrated 10-fold using Kicklab tangential flow filtration cassettes (cut-off 10 kDa, GE Healthcare, Chicago, IL, USA) and adjusted to pH 8.0 in 10 mM Tris before purification in an AKTA Avant system by Steptrap-HP chromatography (GE Healthcare). Final purification was achieved by gel filtration using a Superdex 75 HiLoad 16/60 column (GE Healthcare) equilibrated in phosphate buffered saline without Ca^2+^ and Mg^2+^ (PBS-X). Protein quantity was estimated by peak integration.

### 2.5. IrSPI Refolding

Eight µg of IrSPI were refolded for 30 min at 25 °C in a final volume of 100 µL containing 10 µg of the disulfide isomerase protein (Sigma-Aldrich, St. Louis, MO, USA) with reduced/oxidized glutathione (Sigma-Aldrich) (2 mM/0.2 mM) in borate buffer (100 mM Borate, 150 mM NaCl, pH 7.4).

### 2.6. Circular Dichroism

Circular dichroism experiments were performed in the far-UV region using an Aviv CD 215 model spectrometer equipped with a water-cooled Peltier unit. A 200 µL volume of recombinant IrSPI at 0.075 µg/µL was used to record spectra in a cell with a 1-mm path length in the far-UV range (between 190 to 270 nm) at 25 °C. IrSPI and its cognate buffer were successively screened ten times to produce an averaged spectrum. The spectra were corrected using buffer baselines measured under the same conditions. Data were normalized to residue molar absorption measured in mdeg (M^−1^ cm^−1^) and expressed as delta epsilon (Δε) enabling the BestSel website algorithm to analyze for secondary structure predictions [10].

### 2.7. Analytical Ultracentrifugation (AUC)—Sedimentation Velocity Experiment

The oligomerization state of IrSPI was determined with a sedimentation velocity experiment in a ProteomeLab XL-I analytical ultracentrifuge (Beckman-Coulter, Brea, CA, USA). Briefly, after loading IrSPI at 0.075 mg.ml^−1^ in a 1.2 cm epoxy double-sector cell, the cell assembly was equilibrated for 1.5 h at 20 °C in a four-hole AN60-Ti rotor. The sample was spun at 42,000 rpm for 10 h, and 400 scans for both Rayleigh interference and absorbance at 280 nm were recorded. Absorbance data were collected at a constant radial step size of 0.001 cm. Partial specific volume of 0.70 and the extinction coefficient ε at 280 nm of 14,350 L.mol^−1^.cm^−1^ were theoretically calculated at 20 °C from the IrSPI amino acid sequence using Sednterp software (Spin Analytical, Berwick, ME, USA).

PBS-X density ρ and viscosity η at 1.0053 g.mL^−1^ and 1.018 cP, respectively, were also determined with Sednterp at 20 °C. The sedimentation profile of IrSPI was analyzed in SEDFIT using the continuous size distribution c(s) model [11].

### 2.8. MALDI-TOF/TOF Analysis

Recombinant IrSPI protein purity and integrity were confirmed using a Bruker UltrafeXtreme MALDI-TOF/TOF instrument (Bruker-Daltonics, Germany). Briefly, 1 µL of protein at 0.075 mg/mL was deposited on an MTP 384 ground steel target plate with 1 μL of 2,5-dihydroxybenzoic acid (2,5-DHB) in 50% acetonitrile, 0.1% trifluoroacetic acid as matrix solution. Data were acquired using Flexcontrol software (Bruker-Daltonics, Germany) and shots were recorded in positive ion linear mode. Mass spectra were externally calibrated in the m/z range of 5–20 kDa with a protein calibration standard (Bruker-Daltonics, Germany) and analyzed with the Flexanalysis software (Bruker-Daltonics, Germany). 

### 2.9. Serine Protease Inhibition Assays

Serine protease inhibition assays were performed at 37 °C by using a thermostated Multiskan GO ELISA plate reader (Thermo Scientific). Serine proteases and their corresponding substrates (in brackets) were purchased from Sigma Aldrich and, serine proteases were used with the following concentrations—24.6 nM Trypsin-T1426 (N-Benzoyl-Phe-Val-Arg-*p*-nitroanilide hydrochloride), 46.5 nM Chymotrypsin-C3142 (N-Succinyl-Ala-Ala-Pro-Phe-p-nitroanilide), 21.6 nM Elastase-45125 (N-Succinyl-Ala-Ala-Ala-*p*-nitroanilide), 43 nM Thrombin-T6884 (Sar-Pro-Arg-p-nitroanilide dihydrochloride), and 33 nM Kallikrein-K3627 (N-Benzoyl-Pro-Phe-Arg-p-nitroanilide hydrochloride). Reactions were performed in 20 mM Tris-HCl, 150 mM NaCl, 0.1% BSA, pH 7.4 buffer (except for thrombin inhibition assay performed at pH 6.5), with a final substrate concentration of 0.2 mM. All reactions were performed with both non-refolded and refolded recombinant IrSPI in the refolding buffer, this last buffer being used as a negative control for this latter condition. After initial incubation of serine protease with or without recombinant protein for 15 min at 37 °C, the A_405_ was recorded for 30 min (every 10 s) at the same temperature. Initial slopes of substrate hydrolysis together with their correlation coefficients, were obtained by linear regression analysis using GraphPad Prism software (San Diego, CA, USA). Results were expressed as the percentage inhibition in the presence of IrSPI taking into account the effect of the refolding buffer alone on the enzyme activity.

### 2.10. Ticks Infected with Tick-Borne Pathogens

*B. birtlesii*-infected nymphs pre-fed for three days on mice were obtained from molted larvae infected on mice injected with an in vitro *B. birtlesii* (IBS325^T^) culture as previously described [12]. *B. henselae*-infected females pre-fed for four days were obtained with the previously described membrane feeding system and following the same protocol [5]. *B. henselae* and *B. birtlesii* infection in ticks was confirmed by RT-PCR using primers described by Reis and co-workers [12].

#### 2.10.1. Tick Infection with *Escherichia coli*

*E. coli* (CIP7624 strain, Pasteur Institute, Paris) was cultivated at 37 °C under atmospheric conditions for one night on Luria Bertani (LB)-agar plates and then for 24 h in liquid LB medium. Bacterial concentration was then evaluated by spectrophotometry at 600 nm (OD at 1 represents 2.2 × 10^9^ CFU/mL). Three pathogen-free *I. ricinus* females were nano-injected with 250 nL (1 × 10^7^ CFU) of bacterial suspension into the body cavity using a glass microcapillary attached to a nanoinjector pump (Drummond, Broomall, PA, USA) driven by a Micro 4 controller (World Precise Instruments, Sarasota, FL, USA). Three other females injected with 250 µL of PBS and three un-injected ticks were used as controls. *E. coli* infection was confirmed by qPCR with primers targeting the *WecA* gene [13].

#### 2.10.2. Tick Saliva and Tick Organ Collection

For saliva collection, 34 female *I. ricinus* from the field were pre-fed for five days on a rabbit as previously described [14]. After detachment, ticks were then physically immobilized on microscopy slides with the hypostome positioned in sterile 10 µL tips. After applying 5 µL 5% pilocarpine in methanol to the tick dorsum, ticks were positioned upside down to facilitate collection of saliva and placed in dark, humid chambers as described by Patton et al. [15]. After about 5 h, a pool of approximately 50 µL of saliva was obtained (average of 1.5 µL per tick) and stored at −80 °C until use.

After the saliva collection, salivary gland extracts were prepared. Salivary glands were collected from 17 ticks by dissection under a magnifying glass. Glands were pooled and immediately rinsed three times in cold-PBS (137 mM NaCl, 1.45 mM NaH_2_PO_4_.H_2_O, 20.5 mM Na_2_HPO_4_, pH 7.2). They were then sonicated (Bandelin HD 2070, equipped with the MS 72 probe) three times for 15 s in 1.5 mL of PBS on ice, at 37% power, before concentration to a final volume of 100 µL with the Amicon Ultra filters (3 kDa threshold) and stored at −80 °C until use in western blot. The remaining 17 ticks were then dissected under a magnifying glass, and salivary glands, midgut, ovaries, synganglion and finally the rest of tick body (carcasse) were isolated, rinsed three times in cold-PBS, pooled according to organ and stored at −80 °C until RNA extraction.

#### 2.10.3. Total RNA Extraction

To evaluate gene expression, RNA extraction was performed on eggs, whole unfed larvae, nymphs, adult males and females, fed nymphs from the tick colony, and on tick organs from pre-fed field-collected females using the Nucleo Spin RNA (Macherey-Nagel GmbH & Co. KG, Düren, Germany) kit according to the manufacturer’s instructions. Tissue homogenization was first achieved with Precellys24 (Bertin instruments, Montigny-le-Bretonneux, France) in a lysis buffer containing β-mercaptoethanol with silicate beads for organs and metal beads (2.8 mm of diameter) for whole ticks and eggs. RNA was eluted in 30 µL RNAse free water and RNA concentration was determined by measuring absorbance at 260 nm using a Nanodrop one (Thermo Fisher Scientific).

#### 2.10.4. Quantitative RT-PCR

To evaluate *IrSPI* expression at different tick life stages, RNA samples from one egg deposition, 120 unfed larvae, 20 unfed nymphs, five unfed males, and five unfed females from the tick colony were used. The temporal expression profile of *IrSPI* during feeding was evaluated in RNA samples from 20 unfed, 20 three-day, and 20 five-day fed nymphs from the colony after feeding on rabbits as previously described [14]. To evaluate *IrSPI* expression in different tick organs, pooled RNA samples from salivary glands, midgut, synganglion, ovaries, and carcasses of 17 females from the field that were pre-fed for five days on rabbits were used. Finally, the impact of tick infection with either transmitted or untransmitted bacteria was evaluated. For *B. birtlesii,* pooled salivary gland, gut, and ovary samples from eight infected nymphs pre-fed for three days were used, while the same organs from four uninfected ticks were used as control. For *B. henselae*, pooled salivary glands (13), guts (14), and ovarian tissues (16) from infected females and non-infected controls pre-fed for four days were used. Finally, pooled RNA samples from three *E. coli*-infected females from the colony were analyzed alongside three PBS-injected ticks or three uninjected ticks as controls.

Reverse transcription was performed with SuperScript III (Invitrogen) with 100 to 400 ng RNA per reaction. First-strand cDNA synthesis was performed by using 2.5 µM Oligo dT primers in a total volume of 20 µL at 65 °C for 5 min, 4 °C for 3 min, 50 °C for 50 min, and 70 °C for 15 min. Following RNase H treatment (Invitrogen) at 37 °C for 20 min, expression of *IrSPI, WecA (E. coli), defensin 6*, and *rps4* (a tick housekeeping gene) was evaluated by qPCR using the LightCycler 480 System (Roche, Basel, Switzerland). The SyBr green Master mix (Roche) added to 2 µL of cDNA, contained 6 µL of SyBr green vial 2, and 0.5 µM of both forward and reverse primers in a total volume of 12 µL. Previously described primers were used to detect *IrSPI* transcripts [5], while for *defensin-6,* primers were designed according to the sequence published by Tonk et al. [16]: (F: 5′-TCGTCGTGATGATTGCGGGT-3′ and R: 5′-TCGTCGTGATGATTGCGGGT-3′). The following qPCR cycle conditions were used: 95 °C 5 min; 95 °C 10 s, 60 °C 15 s, 72 °C 15 s, for 45 cycles. Each sample was amplified in triplicate and results were analyzed with Roche LightCycler 480 Software V1.5.0. Relative quantification of gene expression was calculated using the comparative Cp method [17]. *IrSPI* Cp was then normalized using the *I. ricinus*
*rps4* gene [18], and expressed as fold change after ΔΔCp analysis. Analysis of *IrSPI* expression in wild female organs was performed using a plasmid containing *IrSPI* cDNA as a positive control, and *IrSPI* quantity was expressed as *IrSPI* cDNA copy number.

#### 2.10.5. In Situ Hybridization

*In situ* hybridization was performed as previously described [19]. Briefly, a 183-nucleotide long single-stranded DNA probe containing digoxygenin dNTP was prepared via asymmetric PCR (Roche Molecular Biochemicals) and the following primers: 5′-TGACTGAGACACAATGCAGA -3′ (forward) and 5′-TTCCGTACGGACATTCCCGC -3′ (reverse). The salivary glands from partially fed pathogen-free *I. ricinus* females engorged for eight days by means of a membrane feeding system [9] were dissected in ice-cold PBS and subsequently fixed in 4% paraformaldehyde (Sigma-Aldrich P6148) for 2 h at room temperature. After three washes in PBST (PBS + 0.2% Triton), the tissues were treated with 25 µg/mL proteinase K for 10 min at RT. The reaction was stopped by adding PBST-glycine (0.2% of triton and 2 mg/mL glycine) for 5 min. Samples were re-fixed in 4% paraformaldehyde at RT and washed in PBST three times for 5 min. Hybridization was performed with either sense or antisense probe diluted ten times in hybridization solution (HS; 50% formamide, 5x Saline Sodium Citrate solution (SSC), 50 g/mL heparin, 0.1% Tween 20, and 100 g/mL salmon sperm DNA). The samples were then incubated in HS containing the representative probe for 30 h at 48 °C in a humidified chamber. After hybridization, tissues were washed sequentially in HS (12 h, 48 °C), HS-PBST (1:1; 5 min, RT) and PBST (5 min, RT). After blocking in 1% BSA in PBST for 20–30 min at RT, the tissues were immediately incubated with sheep anti-digoxigenin-AP (alkaline phosphatase; Roche Diagnostics, Germany) at a dilution of 1:1000 in PBST overnight at 4 °C. Tissues were washed three times with PBST. Colorimetric reactions were developed by incubation with staining solution made from NBT/BCIP Ready-to-Use Tablets (Roche) according to the manufacturer’s protocol. Salivary glands were mounted in glycerol and examined under an optical microscope (Olympus BX53, Waltham, MA, USA). Digital captured images were assembled and enhanced in Adobe Photoshop CS4.

#### 2.10.6. Anti-IrSPI Serum Production

Anti-IrSPI sera were produced against both the endogenous protein following tick bites in rabbits and the recombinant protein in mice. Two hundred nymphs and around 2000 larvae from our *I. ricinus* colony were fed on one New Zealand white rabbit (Charles River) until feeding completion (seven days) as previously described [14]. At day 17 post-infestation, the rabbit was sacrificed, and blood recovered by cardiac puncture. Five BALB/c mice were used to produce antibodies against IrSPI over a 90-day protocol. Three subcutaneous vaccinations (Days 0, 14, and 28) of 10 µg recombinant IrSPI were performed. Blood samples were collected retro-orbitally at 0, 14, 28, 42, and 56 days after the primary injection, and through final cardiac puncture on day 90. Blood from both mice and rabbits was stored at RT for 2 h before two-step centrifugation at 6500× *g* for 5 min to obtain a clear serum that was stored at −20 °C until used. Recombinant protein immunogenicity and anti-IrSPI murine antibody affinity were confirmed through ELISA experiments.

#### 2.10.7. Western Blot Analysis

A volume of 10 µL of both saliva and salivary gland extracts, as well as 1.5 µg of recombinant IrSPI protein as positive control, were analyzed on 4–15% acrylamide gels under reducing conditions (Mini protean TGX stain-free gels, Bio-Rad, Hercules, CA, USA) followed by UV detection (ChemiDoc, Bio-Rad, Hercules, CA, USA), or were immunoblotted with sera from mice or rabbits after IrSPI immunization or tick infestation, respectively. Transfer to nitrocellulose membranes was performed using the transblot turbo (Bio-Rad) followed by saturation in TBS-5% milk for 1 h at RT and incubation for 1 h at RT with primary antibodies diluted in TBS-5% milk (1/1000 dilution for mice immune serum and 1/100 for rabbit serum). After three washes in TBS-5% milk, antibody binding was detected with the BCIP/NBT Color development system (Roche), using either anti-mouse or anti-rabbit alkaline phosphatase-conjugated antibodies (A3562 or A3687, respectively; Sigma-Aldrich) diluted 1:7500 in TBS-5%. Results were analyzed with the ChemiDoc instrument (Bio-Rad) and the accompanying ImageLab 1.5 software (Bio-Rad).

#### 2.10.8. Thrombin Generation, Clot Waveform, and Fibrinolysis Assays

Thrombin generation assays (TGA) were adapted from the calibrated automated thrombogram (CAT) as described by Jourdi et al. [20]. Briefly, 80 μL calibrated platelet-poor plasma (PPP, Cryopep, Montpellier, France) with or without 0.8 µM refolded IrSPI, and 20 μL PPP-Reagent (Stago, Asnières–sur-Seine, France) were dispensed into wells of flat-bottomed microtiter plates. After 5 min of preheating at 37 °C, thrombin generation was initiated by automated injection of 20 μL of a substrate solution and calcium mixture (FluCa, Stago, Asnières–sur-Seine, France). Progress of Z-Gly-Gly-Arg-7-amino-4-methylcoumarin hydrolysis was recorded every 10 s by a thermostated Tecan Infinite M200 Pro (excitation 340 nm, emission 440 nm). Fluorescence measurements were analyzed using GraphPad Prism software. Lag time, time to peak (TTP), the maximum amount of thrombin formed (peak height), and endogenous thrombin potential (ETP) were computed using the area under the curve method. Clot waveform assays were simultaneously performed by utilizing the Tecan reader’s capacity to simultaneously record fluorescence and A_405_ as described by Jourdi et al. [21]. Data were analyzed using GraphPad Prism software to estimate the lag time of clot formation, defined as the time needed to reach 15% of maximum turbidity. Fibrinolysis was measured by the clot waveform assay except that 6 nM of tPA (Actilyse, Boehringer Ingelheim, Germany) was added to the PPP prior to triggering clot formation. Time to half-lysis was defined as the time needed to halve the maximum turbidity. Obtained results were then compared between positive controls, negative controls (refolding buffer), and samples treated with IrSPI.

#### 2.10.9. IrSPI Impact on Endothelial Cell Apoptosis

The mature human skin microvascular endothelial cell line (HSkMEC) [22], was cultivated in 24-well plates in complete OptiMEM^®^ medium (OptiMEM^®^ with 2% Fetal Bovine Serum, 0.5 µg/mL fungizone, and 40 µg/mL gentamicin, Gibco). After 24 h, the medium was replaced by 1 µM of recombinant IrSPI that had been refolded or not, or by their cognate controls, that is, refolding buffer or complete OptiMEM^®^ medium, respectively. Cells were then incubated for 40 h at 37 °C. At 24 h, staurosporin (1 µM) was added to the corresponding wells as a positive apoptosis control. Cells were detached by trypsinization, washed, labeled with Annexin-V-FITC and propidium iodide according to the manufacturer’s instructions (FITC Annexin V kit and Dead Cell Apoptosis Kit, Invitrogen), and analyzed by cytofluorimetry with the LSR Fortessa X20 instrument (Becton Dickinson, Franklin lakes, NJ, USA).

#### 2.10.10. IrSPI Impact on Host Angiogenesis

The potential cytotoxicity of IrSPI was initially evaluated on HSkMEC cells. After 24 h of culture in 96-well plates in complete OptiMEM^®^ medium, the medium was replaced by refolded or untreated IrSPI at 2, 1, 0.5, 0.25, 0.125, or 0.0625 µM, or by the corresponding controls; namely, refolding buffer or complete OptiMEM^®^ medium. After 48 h of culture, each supernatant was removed and replaced by a solution of Alamar Blue^TM^ (100 µL, at 1/22 dilution in OptiMEM^®^ medium, Invitrogen). Cell toxicity was assessed after 3 h of incubation by measuring fluorescence intensity using the Victor 3 V spectrofluorometric multiwell plate reader (Perkin Elmer, Waltham, MA, USA; excitation at 560 nm; emission at 605 nm). The ability of IrSPI, refolded or not, to modulate host angiogenesis was then evaluated on two human endothelial cell lines, HSkMEC and HEPC-CB1 (human endothelial progenitor cells-cord blood 1), a human progenitor endothelial cell line [23]. To coat plates, 50 µL of Matrigel^TM^ (50% dilution in OptiMEM^®^ medium) was dispensed into 96-well microplates and incubated for 30 min at 37 °C to allow Matrigel^TM^ polymerization. Cells (1.5 × 10^5^ cells/mL) were resuspended in 100 µL OptiMEM^®^ containing either 1 or 0.5 µM of the recombinant protein, refolded or not, or, as controls, the refolding buffer or culture medium, and were then seeded into the Matrigel^TM^ matrix-containing wells. Plates were incubated in a videomicroscope chamber at 37 °C with 5% CO_2_. Pseudo vessel formation was visualized for 9.5 h with an inverted microscope equipped with a CCD camera (Axio Observer Z1, Zeiss, Le Pecq, France), with acquisitions every 30 min via the Zen software (Zeiss, Oberkochen, Germany). Angiogenesis was quantified with the ImageJ software (with Angiogenesis Analyzer) according to several parameters: Number of segments and length, number of meshes, and number of junctions [24].

#### 2.10.11. Splenocyte Proliferation Assay

Spleens of three OF1 mice (8 weeks of age) were gently crushed in individual cell strainers (100 µm). After 2 washes in complete medium (RPMI 1640 (Gibco) with 5% sodium pyruvate, 1% penicillin/streptomycin (Invitrogen), 1% 200 mM L-glutamine (Gibco), and 10% heat-inactivated FCS (Invitrogen), cells were centrifuged for 5 min at 1400 rpm and incubated at 37 °C for 2 min in sterile RBC lysis solution (Invitrogen). Lysis was arrested by adding 30 mL PBS, and cells were pelleted and resuspended in 2 mL complete medium. Cells were then labeled with Carboxy fluorescein succinimidyl ester (CFSE) (10 µM, Molecular Probes) for 5 min at RT in the dark, and quenched by adding five volumes of complete cold medium. CFSE-labeled cells were maintained on ice for 5 min and then washed twice in complete medium. The proliferation of mitogen-stimulated cells was then measured in the presence or absence of IrSPI, in triplicate wells for each condition. For each mouse, CFSE-labeled cells (5 × 10^5^ per well in 50 µL) were dispensed in 96-well U-bottomed plates. Recombinant IrSPI (2 µg per well in 50 µL complete medium) or medium alone was added and cells were incubated at 37 °C in 5% CO_2_ for 2 h. Concanavalin A (ConA) (2 µg/mL in 100 µL of complete medium containing 0.1 mM ß-mercaptoethanol) was then added. Single wells containing CFSE- labeled and unlabeled cells were included for isotype controls and for spectral compensation for CFSE and the viability label. After three days of culture (37 °C in 5% CO_2_), supernatants (150 µL/well) were collected and stored at −80 °C for subsequent analysis of cytokine expression and cells were harvested for phenotypic labeling and flow cytometry.

For phenotypic labeling, all antibodies were purchased from eBiosciences and diluted in PBS containing 0.5% bovine serum albumin and 2 mM EDTA (PBS–BSA–EDTA). To block Fc receptors, cells were resuspended in 50 µL of anti-mouse-CD16/32 (0.5 µg/well) and incubated for 15 min at 4 °C. Cells were then resuspended in a cocktail of PE-anti-CD4 and APC-anti-CD8 (both at 0.125 µg/well) and eFluor 450-anti-CD19 and APC-eFluor 780-anti-CD3e (both at 0.5 µg/well). For isotypic controls, the same concentration of the cognate isotypic antibodies was used. After 30 min, cells were washed and labeled for 30 min with a viability marker (LIVE⁄DEAD^®^ Fixable Aqua Dead Cell Stain Kit, Molecular Probes, Eugene, OR, USA) according to the manufacturer’s recommendations. Cells were washed and fixed with 150 µL of 1% paraformaldehyde in PBS. Flow cytometry was performed using the FACSCanto II cytometer (BD Biosciences, San Jose, CA, USA). Data were acquired and analyzed with FACSDiva (BD Biosciences, San Jose, CA, USA) and FlowJo software (Tree Star, Ashland, OR, USA), respectively. Live cells, distinguished by referring to dot plots of Aqua staining versus side scatter, were gated as B or T lymphocytes on the basis of CD19 or CD3e staining, respectively, and T cells were further distinguished for CD4 or CD8 expression. The CFSE staining of each subset was then examined.

#### 2.10.12. Macrophage Stimulation Assay

Macrophages were isolated from the bone marrow of three 20-week-old BALB/cJRj mice. Mice were sacrificed by cervical dislocation; femurs, and tibias were harvested and kept on ice in PBS-A (without Mg^2+^ and Ca^2+^). Bone marrow was flushed out from cleaned bones without muscle with cold PBS-A using a 10 mL syringe and 25-gauge needle into a sterile 50 mL conical tube placed on ice. Bone marrow cells were centrifuged for 10 min at 4 °C and 300 *g*, the pellet was resuspended in 500 µL of ice-cold PBS-A. Five ml of Gey buffer was added drop by drop to the cell suspension to lyse contaminating red blood cells. After 10 min of incubation at 4 °C, volume was completed to 50 mL with cold PBS-A and the cell suspension centrifuged for 10 min at 4 °C and 300× *g*. The cell pellet was carefully resuspended in 1 mL of complete medium to dissociate aggregates. The cell suspension was diluted in complete medium before cell counting using Malassez microscopy counting slides. Suspensions of 1.5 × 10^6^ cells/mL were cultivated at 37 °C with 7.5% of CO_2_ in Petri dishes (Petri Greiner ref: 664161) using complete DMEM medium (Pan DMEM (Biotech P04-03588) with 15% FCS and 4.5 g/L glucose, 50 µg/mL penicillin and streptomycin, 10 mM HEPES, 50 µM β-ME) supplemented with 50 ng/mL M-CSF-1 (rm M-CSF-1, ref 12343115, ImmunoTools, Friesoythe, Germany). A fresh medium was added after three days of culture. At day six, supernatants were discarded into Falcon tubes, and adherent macrophages were harvested by gentle flushing following a 30 min incubation at 37 °C in PBS-A containing 25 mM EDTA. Individual cell suspensions (1 for each of 3 mice) were then transferred to 12-well plates by using 10 mL of PBS-A at 2 × 10^6^ cells/wells. Four culture conditions were tested for 24 h in complete medium: With or without IrSPI (10 µg/mL), and with or without stimulation cocktail (0.5 µg/mL LPS and 100 U/mL IFN-γ). Respective controls were complete medium or complete medium with stimulation cocktail.

#### 2.10.13. Cytokine Profile Analysis

The impact of IrSPI on cytokine production was evaluated in splenocytes (with or without ConA stimulation) and in LPS and IFN-γ-activated macrophages with the Luminex Procarta PPX-01-S26088EX kit (Invitrogen) targeting mouse Th1 (IL2, IL-1β, IFN-γ, TNFα, IL12p70, IL-18), Th2 (IL4, IL5, IL6, IL-10, IL-13, IL9), Th17 (IL-17, IL-23, IL-27), and Th22 (IL-22) cytokines, chemokines (Eotaxin, Groα-KC, IP10, MIP-1α, MIP-1β, MIP2, MCP1, MCP3, RANTES), and one colony-stimulating factor (GM-CSF), according to the manufacturers’ instructions. Supernatant (50 µL) from splenocytes cultured with or without mitogen stimulation, and with or without IrSPI, and 100 µL of supernatant from activated macrophages cultivated with or without IrSPI, as described above, were used. Each supernatant was incubated for 1.5 h with magnetic beads coated with antibodies directed against the targeted cytokines. The beads with immobilized cytokines were then successively incubated with biotinylated anti-cytokine antibodies and streptavidin-PE, and analyzed using the Luminex^TM^ 200 system (Bio-Rad, Hercules, CA, USA).

#### 2.10.14. Statistical Analysis

GraphPad Prism version 7.8 was used to perform ANCOVA test for inhibition assay linear regression analysis, Student’s *t*-test to evaluate the significance of IrSPI expression and angiogenesis parameters, as well as splenocyte proliferation differences in response to IrSPI exposure. *p*-values < 0.05 were considered as significant (labeled with *), whereas, more significant results with *p*-values < 0.01 were indicated with **.

Cytokine/chemokine Luminex expression analysis was performed for each of the three mice in triplicate, and in duplicate for splenocytes and macrophages. ANOVA analysis was used to evaluate the variability between triplicates or duplicates and between mice for each measured molecule, with and without stimulation. The variable “presence or absence of IrSPI” and the variable mouse were used as a fixed and random effect, respectively. The influence of the protein on the variable “cell response” was expressed as *p* values (*p* < 0.01).

#### 2.10.15. Software

NetNGlyc 1.0 and NetPhos 3.1 servers (DTU Bioinformatics, Lyngby, Denmark) and Phyre 2 software [25] were used in order to predict post-translational modifications and secondary structures of IrSPI, respectively.

## 3. Results

### 3.1. IrSPI Molecular Characterization

The full-length *IrSPI* cDNA encompasses a 342-nucleotide open reading frame encoding a 19-residue signal peptide, followed by a 285-nucleotide sequence encoding a mature 94 amino acid protein. *I. ricinus* and *I. scapularis* sequences show the highest degree of IrSPI identity (Figure 1A). The IrSPI sequence harbours seven cysteines which potentially form three disulfide bonds. As in mammalians, the Kunitz type 1 inhibitors of ticks typically harbor six cysteines, namely, C5, C14, C30, C38 C51 and C55 (according to the original Kunitz numbering system). These cysteines are involved in three intra molecular disulfide bonds: C5-C55, C14-C38 and C30-C51. However, several Kunitz-type inhibitors, including IrSPI, possess an additional cysteine residue, adjacent to the universally conserved C51 (Figure 1B). In these Kunitz, despite additional cysteine residue, the intra molecular disulfide bond pattern for the topologically equivalent cysteines is conserved. The extra cysteine following C51 is unengaged in intra-molecularly bond [26,27]. The putative mature protein has a molecular mass of 8310 Da, a single potential N-glycosylation site, and ten putative phosphorylation sites (Figure 1B). The intact molecular mass of the recombinant IrSPI expressed in Drosophila S2 cells was analyzed by mass spectrometry, using the UltrafleXtreme MALDI-TOF/TOF instrument (Bruker-Daltonics, Germany). Results revealed a 990 Da shift between the experimental mass (13,034 Da) and the theoretical mass (12,044 Da) of the recombinant protein, which is compatible with the addition of ten phosphoryl groups (80 Da) and a single N-linked glycan (161 Da) as previously hypothesized via NetNGlyc 1.0 and NetPhos 3.1 servers (Appendix A).

The predicted protein structure, as determined via Phyre 2 software, consists of two β-sheets, two α-helices, and a single Kunitz-like inhibitory domain, suggesting that IrSPI belongs to the Kunitz type I protease inhibitor family (Figure 1C). Recombinant IrSPI protein was used to confirm this structure, but circular dichroism experiments failed to reveal the expected α-helix signals (Appendix A). In addition, the main population observed via analytical ultracentrifugation had a sedimentation coefficient of 1.6S and a frictional ratio (f/f0) of 1.2, consistent with a monomeric and globular form of IrSPI (Appendix A).

The absence of helices in the recombinant IrSPI structure may prevent disulfide bond formation, consequently causing hydrophobic residues to be exposed at the protein surface. Thus, the globular aspect of IrSPI could be due to an abortive folding process, and its globular form might represent a folding intermediary and not the correct functional IrSPI structure. Altogether, results suggest that the recombinant protein may be misfolded, which might compromise its functional activity as a protease inhibitor. Consequently, serine protease inhibition assays were performed with a refolded form of recombinant IrSPI, and the refolding buffer was used as a control.

### 3.2. Serine Protease Inhibition

Since the P1 residue of IrSPI is an alanine, we hypothesized that it could inhibit serine proteases of the elastase-like family. We thus, assessed the ability of IrSPI to inhibit cleavage of the respective substrates of several serine proteases, including elastase. Although the non-refolded recombinant protein did not inhibit any serine protease; refolded IrSPI in excess (1 µM) moderately, but significantly decreased elastase catalytic activity (23.9%; ANCOVA test, *p* = 0.0002), and that of chymotrypsin very slightly (1.7%) (Figure 2). Trypsin, thrombin, and kallikrein were not inhibited.

### 3.3. IrSPI Expression and Localisation

To explore IrSPI function, we analyzed its expression by quantitative RT-PCR in all tick stages from our pathogen-free *I. ricinus* colony. IrSPI mRNA was not detected in eggs, unfed larvae, nymphs, adults of either sex or in fed larvae on days 1 and 3 of engorgement. However, comparison of transcript abundance in unfed, 3-day fed, and 5-day fully engorged nymphs revealed significant upregulation of IrSPI mRNA following five days of feeding (Figure 3A). IrSPI mRNA expression was then evaluated in different tissues in a pool of 17 field-collected females pre-fed for five days on a rabbit that was shown to be infected with *Babesia venatorum* and *Rickettsia helvetica* by PCR as previously described [8]. Results showed that in adults, and at roughly halfway through feeding, *IrSPI* is expressed in SG, guts, synganglion, ovaries and the rest of tick bodies (carcasses), with significantly higher expression in SG (Figure 3B). *IrSPI*-specific transcripts were then analyzed in various organs of *B. henselae*-infected *I. ricinus.* Results confirmed overexpression in SG following infection, and also revealed upregulation in the gut, and in ovaries to a lesser extent (Figure 3C). Such overexpression was not found following infection with *B. birtlesii*, a phylogenetically proximal bacterium.

*IrSPI* expression was also analyzed after infecting ticks with a non-transmitted bacterium, *Escherichia coli*, to evaluate the protein’s potential implication in tick innate immune responses. The efficiency of this response was validated in parallel by evaluating the expression of *defensin6*, which is known to be upregulated in ticks following *E. coli* infection [16]. While we confirmed that *defensin6* was upregulated following infection, expression of *IrSPI* remained unchanged (Figure 3D).

To document *IrSPI* expression in the salivary glands and to determine whether the gene product is present in tick saliva injected into the vertebrate host, *IrSPI* mRNAs were analyzed by in situ hybridization in salivary glands of pathogen-free, partially-fed females. The *IrSPI* transcript was identified in the secretion vesicles of type II acini, which are responsible for saliva secretion in hard ticks [28] (Figure 3E).

The presence of IrSPI protein was then evaluated in both tick saliva and in salivary gland extracts obtained from a pool of 17 *I. ricinus* females collected from the field, then pre-fed on a rabbit for five days. SDS-PAGE analysis revealed major differences in the protein composition of saliva and salivary gland extracts, with higher total protein observed in salivary glands (Figure 4A). The presence of IrSPI was confirmed by western blot analysis using antiserum from mice immunized with recombinant IrSPI (Figure 4B), as well as serum collected from an experimentally tick-infested rabbit (Figure 4C). Native IrSPI was only faintly detected by the mouse antiserum in tick salivary glands, and not at all in saliva (Figure 4B). However, the recombinant counterpart was detected, although weakly, by the serum from a tick-infested rabbit, demonstrating that IrSPI generates an antibody response in rabbits, and is indeed injected into the host by ticks via the saliva during blood-feeding (Figure 4C). These results suggest possible differences regarding immunogenicity/conformation and/or protein quantity between the recombinant and the native proteins, since mouse antiserum recognized lower levels of native versus recombinant IrSPI.

### 3.4. Anticoagulant Activity

Thrombin generation and clot waveform assays were used to test whether IrSPI displays anticoagulant activity. Thrombin formation was triggered by adding calcium to a premix of plasma, phospholipid vesicles, and tissue factor, in the presence or absence of varying amounts of ‘refolded’ IrSPI. The kinetic analysis suggested that IrSPI had little or no impact on clot formation (Figure 5). At concentrations of up to 0.8 µM IrSPI, clot formation lag-time and amounts of thrombin formed were comparable between the positive control (peak at 371 s) and samples containing IrSPI (peak at 371 s) or refolding buffer alone (peak at 357 s, Figure 5A). IrSPI also had no significant impact on fibrin formation, with respective clotting-times of 105 s for controls (with or without refolding buffer) versus 119 s for samples containing IrSPI (Figure 5B). IrSPI’s clot lysing ability was also investigated to evaluate its effect on clot stability. Fibrin formation was triggered as above—with a mixture of phospholipid vesicles, tissue factor, and calcium, but in the presence of tissue plasminogen activator (tPA) to induce fibrin clot lysis. Again, there was no significant difference between half-lysis times using refolding buffer with or without IrSPI (279 versus 297 s, respectively; Figure 5C).

### 3.5. Apoptosis and Angiogenesis

As IrSPI is known to be a component of tick saliva, which precludes tissue remodeling and angiogenesis, we next hypothesized that it could modulate the formation of new vessels by promoting apoptosis and/or by inhibiting microvascular endothelial cell proliferation. The ability of refolded or misfolded recombinant IrSPI to induce apoptosis in comparison with cognate media controls was first evaluated in HSkMEC endothelial cells by flow cytometry cell viability and apoptosis markers, i.e., cells labeled with Annexin V-FITC, but not propidium iodide corresponded to apoptotic cells (Figure 6). Adding 1 µM IrSPI (refolded or not) to culture medium had only a weak effect (11.9% and 9.2% apoptotic cells versus 10.5% with culture medium alone); the greatest effect was actually obtained by the addition of refolding buffer without IrSPI (25.5%).

Two endothelial cell lines were used to investigate the impact of IrSPI on angiogenesis: HSkMEC and HEPC-CB1. We first evaluated IrSPI toxicity (with dilutions between 0.0625 and 2 µM, refolded or not), as well as culture—or refolding-media toxicity on HSkMEC cells. Concentrations of refolded IrSPI (0.5 µM and 1 µM) that did not induce significant toxicity compared to control (mortality of 18% and 29%, respectively), were used to evaluate IrSPI’s impact on angiogenesis. Compared with culture medium, non-refolded IrSPI at either concentration had no effect on the number of junctions, master junctions, segments, master segments, meshes, or total segment lengths in either of the two endothelial cell lines (Figure 7). In both cell lines, both concentrations of refolded IrSPI reduced several angiogenesis parameters, but this effect was mainly due to the refolding buffer and not IrSPI itself (Figure 7).

### 3.6. Splenocyte Proliferation Assay

Next, we evaluated whether IrSPI might play a role in host immunomodulation during tick feeding. Splenocytes isolated from three OF1 mice were cultured in the presence or absence of the mitogen concanavalin A (ConA) for three days. The impact of non-refolded IrSPI on the proliferation of three lymphocyte subsets (CD4^+^ and CD8^+^ T cells, B cells) was then evaluated by CFSE dye dilution assay via flow cytometry. In the absence of IrSPI, mitogen-stimulated T, but not B cells, displayed robust proliferation, though with some inter-individual variability probably attributable to the use of outbred mice (Figure 8).

In the presence of IrSPI, the proliferation of the global T cell population diminished, as evidenced by increased frequencies of undivided cells and decreased frequencies of cells that had undergone multiple divisions. Much of the observed reduction in proliferative capacity could be attributed to the CD4^+^ T cell subset; indeed, in the presence of IrSPI, the diminution in the frequency of cells having undergone one or more divisions was statistically significant in CD4^+^ (18%), but not in CD8^+^ (3.6%) T cells. Thus, IrSPI diminished the capacity of T cells, and especially that of the CD4^+^ subset, to respond to mitogenic stimulus, thus, suggesting a role for IrSPI in immune modulation.

### 3.7. Cytokine Expression in Response to IrSPI

In order to gain insight into how IrSPI might influence immune cells, cytokines and chemokines secreted into the supernatants of mitogen-stimulated or unstimulated splenocytes were compared using a multiplex bead-based assay. The 26-plex panel included 17 cytokines representing multiple T helper subsets (Th1, Th2, Th9, Th17, Th22, and Treg) and nine chemokines. In the absence of mitogen stimulation, but in the presence of IrSPI, the secretion of several cytokines and chemokines was significantly diminished (*p* < 0.01), in particular IP10 (−32.8%), MCP3 (−72.3%), MIP-1β (−26%), and RANTES (−15.6%) (Table 1 and Appendix A). IrSPI also significantly decreased several chemokines in mitogen-stimulated splenocytes including IP10 (−46.5%), MIP-1β (−17.9%), RANTES (−37.9%) and Eotaxin (−21.3%) (*p* < 0.01). Moreover, expression of several cytokines, namely, IFN-γ (−50.3%), IL-1β (−45%), IL-18 (−46%), IL-13 (−75.1%), IL-6 (−54.8%), TNF-α (−46,1%), IL-9 (−48.8%), and GM-CSF (−71.9%) was also significantly reduced (*p* < 0.01), (Table 1 and Appendix A). In contrast, a single cytokine, IL-2, was significantly upregulated (+49.2%) following IrSPI exposure.

To address the impact of IrSPI on sentinel immune cells, we also compared T helper cytokines and chemokines in the supernatants of IFN-γ-and LPS-activated murine macrophages grown in the presence or absence of IrSPI. All tested T helper cytokines and chemokines were diminished, but only the decrease in IL-5 was statistically significant (−9.6%; *p* < 0.01; Table 1 and Appendix A).

Taken together, our data suggest that, apart from IL-2, IrSPI repressed splenocyte—and macrophage-activated secretion of multiple T helper cytokines and chemokines, consistent with a role for IrSPI as a modulator of the host immune response against tick bites.

## 4. Discussion

Further knowledge is continually accumulating regarding the capacity of hard and soft tick saliva to counteract native host immune responses to enable successful blood meal completion [29,30,31]. Relatively few salivary compounds, however, have undergone thorough molecular characterization, or have been identified as the active factors responsible for this process (recently reviewed in [3]). We previously identified a tick serine protease inhibitor—IrSPI—potentially present in *I. ricinus* saliva and likely implicated in tick feeding and in *I. ricinus* salivary gland infection by *Bartonella* [5]. The presence of genes homologous/paralogous to *IrSPI* in both *I. ricinus* and *I. scapularis* ticks suggests that it belongs to a family of SPI proteins that have important physiological functions, with highly conserved regulatory roles. Here we also provide evidence that IrSPI is a Kunitz SPI that is induced during blood-feeding and appears to be present in tick saliva, even if this will have to be validated by proteomic studies. Moreover, we show that IrSPI inhibits elastase and modulates host immune responses. It is likely that such immunomodulation substantially contributes to successful feeding and engorgement of *I. ricinus* ticks, and thereby facilitates TBP transmission.

Production of functionally active recombinant protein is known to be difficult. As an example, the *Dermacentor andersoni* p36 immunosuppressive salivary gland protein was only active when the recombinant protein was expressed in insect cells, but not in bacteria [32], suggesting that functional activity might depend upon proper post-translational modifications, such as glycosylation. Similarly, Konnai and co-workers have tentatively attributed the inability of *Ixodes persulcatus* lipocalin to bind histamine to the expression of recombinant protein in *E. coli* [33]. In the present study, we expressed IrSPI in *Drosophila* cells and showed that the protein was functional, e.g., it exerted immunomodulatory activity. However, circular dichroism analysis of recombinant IrSPI suggested that it lacked important Kunitz secondary structural features. It is likely that the typical disulfide bonds pattern of kunitz type 1 (C5-C55, C14-C38 and C30-C51) was not well-formed, and thus, the recombinant protein might be jam in a folding intermediate. That prevents the good exposition of the reactive loop and the inhibition of serine proteases. Nonetheless, when subjected to a refolding step that consists of reforming good disulfide bonds intramolecularly thanks to the isomerase activity of PDI, the recombinant protein displayed inhibitory activity against elastase.

In light of its primary structure, IrSPI belongs to the Kunitz SPI superfamily, whose members competitively inhibit serine protease activity in a reversible lock-and-key fashion. In ticks, several members of this superfamily are known to subvert host defense mechanisms during feeding [7]. Invertebrate Kunitz inhibitors possess diverse inhibitory effects on several serine proteases depending on their amino acid sequence, especially the P1 residue of the reactive site loop. Positively charged Lys or Arg, as P1 residues are associated with inhibition of trypsin-like serine proteases; large hydrophobic residues, such as Phe, Tyr, or Leu, with chymotrypsin-like serine proteases; and small hydrophobic residues, such as Ala or Val, with elastase-like serine proteases [34]. We showed that IrSPI, though harboring Ala as its P1 residue, does indeed inhibit elastase albeit to a lesser extent than that reported in the literature for other tick kunitz protease inhibitors [35,36]. IrSPI also inhibited chymotrypsin activity, albeit weakly, in keeping with the partial fit of an Ala-bearing SPI within the protease active site, but less effectively than an SPI bearing large hydrophobic amino acids at this P1 position. Elastase, a serine protease synthesized and released by neutrophil cells following trauma, is responsible for tissue inflammation and remodeling [37]; chymotrypsin also participates in this latter process [38]. Through its inhibitory activity, IrSPI might, thus, reduce inflammation and inhibit tissue repair processes following the tick bite, as has been suggested for other Kunitz SPIs, such as *Ixodes scapularis* tryptogalinin which also inhibits elastase, among other proteases [39].

We showed that *IrSPI* mRNA was expressed in various organs from pooled ticks, including SG, gut, ovaries, and synganglion, as well as the carcasses of wild pre-fed ticks that harbored—for at least one of the pooled ticks—two TBP. This broad tissue distribution suggests that IrSPI might be implicated in multiple aspects of tick biology. *IrSPI* expression in pools containing infected ticks is compatible with its induction by TBP, as we previously demonstrated for *B. henselae* [5]. In fact, TBP infection has already been shown to modulate the expression of many tick genes, and a number of these—including some encoding Kunitz proteins—have been identified as implicated in TBP transmission [40]. Such molecular interactions reflect the vector competence of an arthropod [4], and may be TBP-specific, as shown for IrSPI regarding *B. henselae* and *B. birtlesii*. Previous RNAi experiments have also shown that IrSPI is likely involved in bacterial adhesion, invasion, and/or multiplication within tick salivary glands, and also potentially in tick defense mechanisms, as it may act against other bacteria in competition with *B. henselae* [5]. This latter hypothesis was, thus, evaluated using *E. coli,* but *IrSPI* upregulation was not observed following tick infection with this non-transmitted bacterium.

We next demonstrated that *IrSPI* expression is induced in tick salivary glands by the blood-feeding process, both in nymphs at the end of engorgement (day 5) and in females in the middle of the feeding process, and in the latter context on day 5 on rabbits, and on days 4 and 8 on membrane [5]. The immunomodulatory activity of IrSPI that we observed—that is, reduction in T cell proliferation and more particularly in proliferation of the CD4^+^ T subset—is compatible with late expression of IrSPI, as the T-cell response is only expected to be initiated following the early inflammatory response. Similar blood-meal induction has also been reported for the *Irsgmg-150466* transcript, which likely corresponds to the *IrSPI* gene as it has 100% identity at the protein level, but without a signal peptide sequence. *Irsgmg-150466* expression peaked 12 h after attachment, but then decreased after 24 h and 12 h in *I. ricinus* female and nymph salivary glands, respectively [41]. Chmelar and co-workers reported the expression of a homologous gene (contig 240) in *I. ricinus* female salivary glands 24 h after tick attachment [42]. Dynamic expression of IrSPI paralogs was also recently reported by Dai et al. in *I. scapularis* [43], lending support to variable BPTI/Kunitz gene expression by both *I. ricinus* and *I. scapularis* during blood meal ingestion, with time-dependent up- or down-regulation. Nevertheless, it should be noted that in these studies, expression level is only based on mRNA detection, which may differ from protein expression, as variable transcript expression does not necessarily translate into variation in expression of the corresponding protein.

The presence of a signal peptide combined with salivary gland transcript expression indicated that IrSPI was likely to be a secreted protein present in tick saliva. We have shown that *IrSPI* mRNA is expressed in type 2 acini, which are responsible for saliva secretion, and using anti-IrSPI antibodies raised in tick-infested rabbits, that native IrSPI protein may be present in tick saliva and injected into the host during feeding. This demonstrates that IrSPI takes part in the molecular dialogue between the tick and its vertebrate host, and strongly argues in favor of a role for IrSPI in the tick feeding process.

Host skin is the first barrier that ticks need to overcome for successful feeding. Upon tissue damage, host skin cellular components—including epithelial cells, fibroblasts, and endothelial cells—generate biomolecules that facilitate tissue remodeling and wound healing, such as angiogenesis. In order to remain attached to their host and capable of obtaining blood over several days, these slow-feeding hematophagous arthropods must inhibit these processes. The first study demonstrating that tick saliva inhibits angiogenesis was published in 2005 [30]. Since then, few salivary molecules have been reported to modulate angiogenesis and wound healing. These include Kunitz SPI from *Haemaphysalis longicornis* (Haemangin [44]), *R. microplus* (BmTI-A [36], and BmCI [45]), and *Amblyomma cajennense* (Amblyomin-X [46]). Nevertheless, although IrSPI, like BmTI-A and BmCI, acts as an elastase inhibitor, it does not seem to exert direct activity on apoptosis of HSkMEC cells, or angiogenesis in HSkMEC and HEPC-CB [1] cells. It should be noted, however, that this in vitro observation may not hold true for the whole organism, where suppression of angiogenesis-related proinflammatory cytokines by IrSPI, as observed in both splenocytes and macrophage cells, might exert an indirect effect.

To initiate blood-feeding, the tick must lacerate the host tissue, which exposes to blood large quantities of tissue factors triggering activation of the blood coagulation cascade that induces fibrin clot formation and its subsequent lysis. Blood coagulation is, thus, rapidly triggered following tick bite-induced blood vessel damage. Several tick factors counteract vertebrate blood hemostasis (reviewed in [3]), the majority belonging to the SPI family. In fact, most of the tick SPI injected into the host during blood-feeding facilitate completion of the long blood meal by targeting one or several serine protease(s) involved in blood coagulation pathways, thereby preventing fibrin clot formation and maintaining blood in a fluid state at the tick bite site [7]. IrSPI, however, did not appear to modulate blood coagulation, as no effect was observed on thrombin generation or fibrin formation. In addition, albeit several tick SPI accelerate lysis of fibrin clot [7], IrSPI failed to demonstrate such activity. Nevertheless, this result was somehow expected given that the P1 residue of IrSPI is alanine, suggesting elastase-like targets rather than blood coagulation proteases which are all trypsin-like enzymes [47]. Further investigations would be required to determine whether IrSPI regulate other blood coagulation processes, such as platelet aggregation.

To avoid host rejection, ticks have developed a complex armament to circumvent both innate and adaptive immune responses, as well as complement activation [29]. Relatively few tick salivary factors responsible for this immunomodulation have been identified (reviewed in [3]). These factors have been reported to play a role in the activation and/or recruitment of multiple resident and infiltrating cell types at the tick bite site, including macrophages, mast cells, dendritic cells, endothelial cells, keratinocytes, and lymphocytes, as well as in the polarization of an initial Th1 helper response towards a Th2 orientation, considered to be more favourable to tick blood-feeding [29]. Here we showed that IrSPI inhibits mitogen-induced proliferation of murine CD4^+^ T cells, as has been reported for saliva of not only *I. ricinus* [48], but also several other tick species including *D. andersoni* [49], *A. variegatum* [50], and *I. dammini* [51]. It is noteworthy that in our study, IrSPI inhibited secretion of IL-9, which promotes the proliferation of CD4^+^ rather than CD8^+^ T cells [52]. Until now, a small number of salivary proteins with immunosuppressive capacity have been characterized, including Salp15 from *I. scapularis* [53], the most highly-investigated protein, P36 from *D. andersoni* [32], and Iris from *I. ricinus* [54]. It should be noted that in our study, the impact of IrSPI was assessed on mouse splenocytes, which comprise, in addition to the high proportion of T cells, multiple other immune cell types, such as B cells, macrophages, dendritic cells, eosinophils, basophils, natural killer cells and neutrophils. Thus, indirect actions of IrSPI on T cells via other cells should not be excluded, as has already been suggested for dendritic cells [55]. Indeed, we observed that IrSPI decreased splenocyte expression of multiple chemokines and cytokines that directly or indirectly stimulate T cell proliferation, including MIP-1β, RANTES, and IL-9 [56,57]. Furthermore, we showed that IrSPI also repressed the expression of IL-13, which is mainly produced by Th2 lymphocytes, and which may regulate proliferation and differentiation of DCs and B cells [58].

At the tick bite site, and in response to tissue damage, numerous cell types (including fibroblasts, endothelial cells, macrophages, DCs, T cells, neutrophils, eosinophils, and resident tissue cells) are able to secrete immunological factors, thereby promoting inflammation, tissue remodelling, wound healing, angiogenesis, and pathogen clearance. To repress immune cell activation and proliferation that, if unchecked, would compromise tick feeding, tick saliva, as a rich source of immune modulators (reviewed in [3]) acting in concert to achieve a stronger effect, hijacks cellular immune responses and notably secretion of cytokines and chemokines, thus, affecting the previously mentioned processes. Our results showed that for *I. ricinus*, IrSPI, as part of this salivary cocktail, represents one of the molecules that are likely to contribute to the modification of these molecular profiles, and suggest a broader function of IrSPI along with tick feeding in vivo due to serine protease inhibitor numerous roles in mammalian physiology.

Inhibition of proinflammatory molecules by total salivary gland extracts of *I. ricinus* has already been reported [59]. Among IrSPI down-regulated molecules, MIP-1β, GM-CSF, and RANTES—strongly associated with secretion of IFN-γ—are considered to be proinflammatory factors typically expressed in response to infection, promoting the recruitment of effector cells to the site of pathogen entry [60]. Our data also showed that IrSPI decreased the expression of IFNγ as has been reported for salivary gland extracts of *I. ricinus* [61]. IFNγ is principally produced by cytotoxic and Th1 cells, and constitutes one of the primary regulators of both innate and adaptive immunity, as well as inflammation. Local tissue inflammation and allergic reactions at the tick bite site are also controlled by IrSPI through the inhibition of eotaxin-1, IL-18, or IL-9. Inhibition of this last cytokine has also been reported for the salivary cysteine protease inhibitor sialostatin L from *I. scapularis*, thus, preventing the development of experimental asthma [62].

In addition, and like the dendritic cell modulator Japanin from *Rhipicephalus appendiculatus* [63], IrSPI also repressed the expression of TNFα, which is predominantly produced by both activated macrophages and T lymphocytes in inflammatory and infectious conditions. Although unable to drive the differentiation of naïve CD4^+^ cells towards a Th2 phenotype in a direct manner, IL-13, also repressed by IrSPI, may affect T cell function indirectly via downregulation of proinflammatory Th1 cytokines, including IL-12, which have been shown to be also repressed by OmC2 from the soft tick *Ornithodoros moubata* [64].

Like Japanin [63], IrSPI decreased IL-1β. In addition to triggering the release of other proinflammatory cytokines, this cytokine—which is secreted following injury—activates neutrophil and macrophage phagocytosis and the release of toxic oxygen and nitrogen radicals. As serine proteases from neutrophils and macrophages, such as elastase, have been reported to process pro-IL-1β into active fragments [65], we, thus, cannot exclude a role for IrSPI as an indirect inhibitor of IL-1β secretion through inhibition of these molecules. As this cytokine was among the most highly downregulated in the presence of IrSPI, and given the importance of IL-1β in both immune cell recruitment and nociception, its suppression by IrSPI may well participate in avoidance of tick rejection by the host. Besides IL-1β pain sensation is reinforced by the release of pro nociceptive chemokines and cytokines, including TNFα, IFNγ, IL-6, IL-18, MCP3, RANTES, and MIP-1β, at the tick bite site [66], and these are also repressed by IrSPI in cultured splenocytes. Immune cell recruitment might also be impacted by IrSPI through the modulation of MIP-1β, RANTES, GM-CSF, and IP10 or IL-6, triggering the expression of T cell attractant chemokines and eotaxin-1, TNF-α, or IL-13 which regulate cell adhesion proteins [67].

As ticks must maintain blood flow at the bite site in order to feed successfully, inhibition of several cytokines/chemokines by IrSPI might antagonize host homeostasis and angiogenesis. Affected cytokines include RANTES, which modulates neovascularization, endothelial cell migration and spreading [68], IL-6, which induces VEGF [69], TNF which stimulates *COX 2* [70], and eotaxin-1 which affects endothelial cell chemotaxis and in vivo blood vessel formation [71].

The immunosuppressive effect of IrSPI on the host will naturally also influence the transmission of tick-borne pathogens. Several of the cytokines/chemokines repressed by IrSPI, including RANTES, MIP-1β, IL-18 and IL-9, are also implicated in pathogen control and clearance mechanisms, including effector cell recruitment to the pathogen entry site [60,72]. While IrSPI significantly inhibits most proinflammatory cytokines and chemokines, a single cytokine, IL-2, was upregulated. This result was somewhat unexpected in the light of published data reporting decreased IL-2 expression by tick saliva [29]. Nevertheless, while IL-2 is often considered to be proinflammatory, it can also behave as an anti-inflammatory cytokine by promoting differentiation of Treg cells, as well as their long-term survival, and suppressive activities [73], thereby decreasing local inflammation and promoting pathogen tolerance [74].

Together with the splenocyte results, the macrophage response to IrSPI confirms that this salivary molecule downregulates both proinflammatory and Th1-promoting cytokines. A similar result has been reported where TNF-α, IL-12, and IFN-γ were decreased in LPS-activated bovine macrophages exposed to *R. microplus* saliva [75]. While all tested cytokines decreased in this study, only IL-5 from mouse macrophages was significantly lowered. This cytokine is reported to have proinflammatory functions, including eosinophil activation, B cell growth, and the stimulation of antibody production. Its proinflammatory features are reinforced via its ability to increase the susceptibility of B and T cells to IL-2, and subsequently the production of cytotoxic T cells. Although this cytokine is predominantly expressed by CD4^+^ T cells, recent evidence suggests that macrophages may also produce IL-5 [76].

## 5. Conclusions

IrSPI is produced by the most common tick vector in Europe and seems to be a vital component of the tick salivary cocktail that enables the tick to overcome the host immune system’s defenses to feed successfully and transmit pathogens. While further investigations are required to gain a full understanding of the molecular processes that modulate the host immune response, this work represents a significant advance in the characterization of key immunomodulatory molecules in tick saliva. IrSPI also represents a potential target antigen for anti-tick vaccine development, as host immune responses elicited against such immunomodulatory tick molecules may diminish tick infestation and/or TBD transmission. Anti-tick vaccines hold the promise of affording broad protection against multiple TBD transmitted by the same vector. The utilization of “exposed” salivary antigens, such as IrSPI, rather than “concealed” tick antigens to which the host is never naturally exposed, may enable the host response to be naturally boosted upon exposure to ticks [77]. Also, immunity against such proteins may block the feeding process prior to pathogen transmission, which typically occurs many hours or even days after tick attachment. Finally, and although we have reported an impact of *IrSPI* silencing on both tick feeding and bacteria transmission, IrSPI belongs to a protein family with extensive functional redundancy [78]. In consequence, the vaccinal potential of IrSPI would likely be evaluated by promoting immune responses against epitopes that are common to the IrSPI protein family. Moreover, the protection afforded by IrSPI should ideally be assessed in the context of multivalent anti-tick vaccines comprising a cocktail of salivary antigens involved in distinct mechanisms that counteract host defenses by different mechanisms. Furthermore, due to its immunomodulatory activities, IrSPI may also represent a potential candidate as a therapeutic molecule in pathologies linked to immune cell proliferation disorders.

## Figures and Tables

**Figure 1 vaccines-07-00148-f001:**
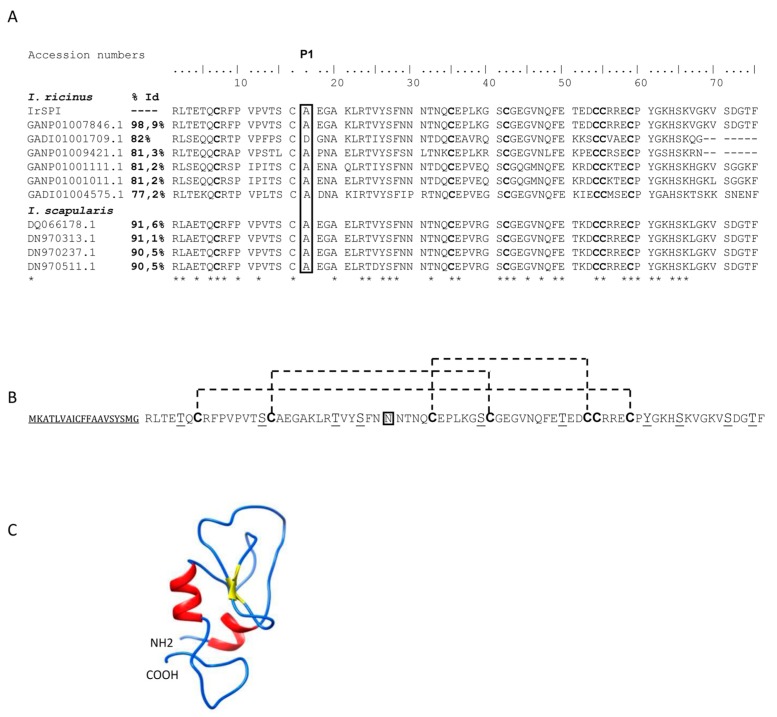
Structures of *Ixodes ricinus* serine protease inhibitor (IrSPI) (KF 531922.2) and amino acid sequence alignment of *I. ricinus* and *I. scapularis* Kunitz protease inhibitors. (**A**) Amino acid sequence alignment of IrSPI with other known *I. ricinus* and *I. scapularis* Kunitz protease inhibitors with corresponding accession numbers. Identical amino acids are marked with asterisks, and the P1 residue is highlighted by a rectangle. Accession numbers in GenBank are indicated for each gene, as well as the percentage of identity with IrSPI (% Id) (**B**). Primary and (**C**) secondary IrSPI structures. The signal peptide is underlined. Post-translational modifications are predicted to include 10 putative phosphorylation sites (underlined) and one putative N-glycosylation site (double underlined). The seven cysteine residues (in bold) are capable of forming three disulfide bonds (dashed line) and, combined with the formation of 2 β–sheets (yellow) and 2 α-helices (red), bestowing to the IrSPI protein its unique Kunitz/BPTI fold.

**Figure 2 vaccines-07-00148-f002:**
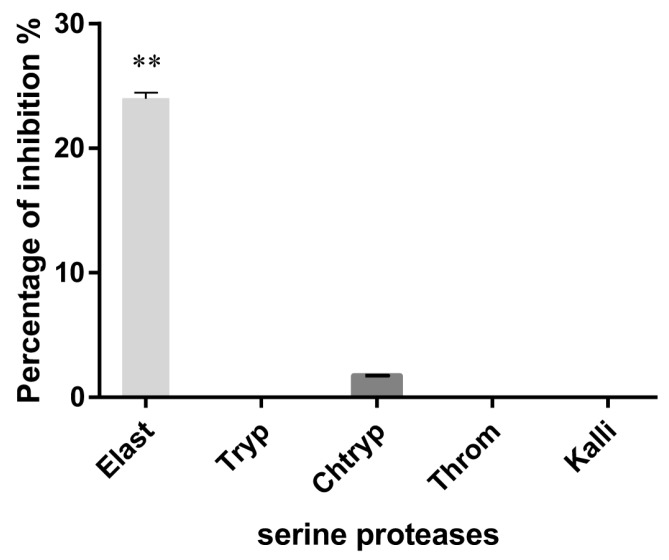
Inhibition of serine proteases by IrSPI. The ability of IrSPI (1µM of refolded protein in the refolding buffer) to inhibit the activity of elastase (Elast), trypsin (Tryp), chymotrypsin (Chtryp), thrombin (Throm), and kallikrein (Kalli) on synthetic substrates was evaluated at 37 °C for 30 min. Enzyme activity was evaluated from the slope of A_405_ versus time and results were expressed as the mean of IrSPI-induced percentage inhibition after subtraction of the refolding buffer effect and comparison with the control. Double asterisks (**) indicate significant results as determined via ANCOVA analysis (*p* < 0.001).

**Figure 3 vaccines-07-00148-f003:**
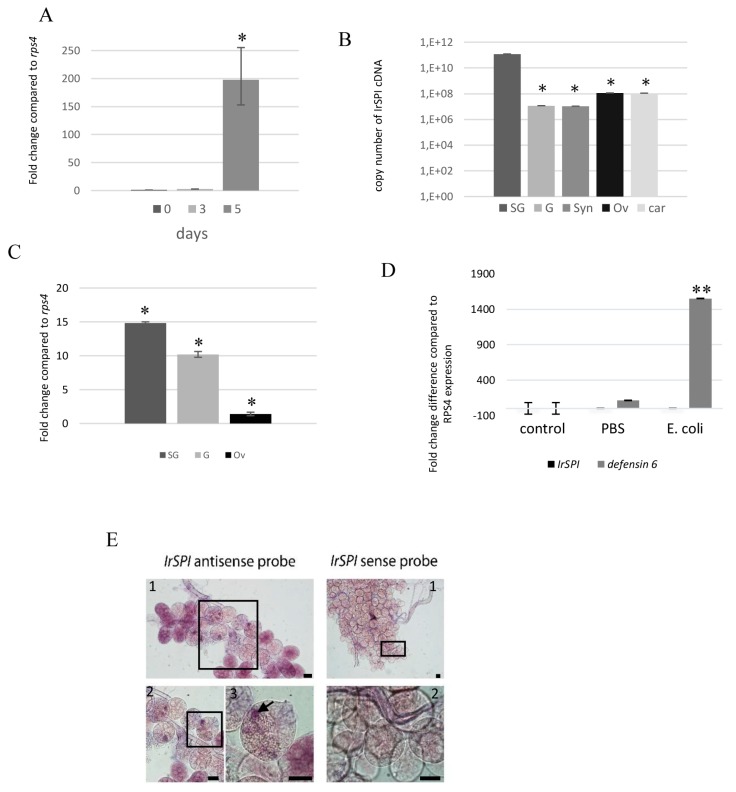
*IrSPI* transcript profiling and localization. (**A**) Relative *IrSPI* expression in pathogen-free nymphs fed on rabbits revealing the delayed gene expression on day 5. At each feeding time point (Days 0, 3, and 5), 20 pathogen-free nymphs were used, and *IrSPI* expression was assessed by quantitative RT-PCR relative to the *rps4* gene. (**B**) Relative *IrSPI* expression in several organs of wild female ticks pre-fed for five days on the rabbit. Pools of salivary glands (SG), gut (G), synganglion (Syn), ovaries (Ov), and the remaining body, i.e., the carcasses (car) from 17 females were used for each condition, and *IrSPI* expression was assessed by quantitative RT-PCR according to the copy number of *IrSPI* plasmid clones. (**C**) Relative expression of *IrSPI* in pools of 13 salivary glands (SG), 14 guts (G), and 16 ovaries (Ov) from *B. henselae*-infected females pre-fed for four days assessed by quantitative RT-PCR, compared with uninfected samples, and expressed as a fold change with respect to the *rps4* gene. (**D**) Relative *IrSPI* and *defensin6* expression after *E. coli* injection. Three pathogen-free females per group either received no injection (Control), 250 nl of PBS (PBS), or 250 nl of 10^7^ bacterial CFU (*E. coli*). After 24 h incubation, quantitative RT-PCR was used to assess the mean expression of *IrSPI* and *defensin 6* normalized to *rps4*. (**E**) In situ hybridization of *IrSPI* mRNA in salivary glands of pathogen-free females pre-fed for eight days on the membrane. A specific *IrSPI* anti-sense probe was incubated on salivary glands and visualized through the addition of NBT/BCIP. Arrows indicate the presence of *IrSPI* transcripts in cells containing secretory vesicles in type II acini. An *IrSPI* sense probe was used as a negative control. Fields 1 to 3 are successive zooms. The scale bars are 60 µm. One asterisk (*) indicate significant (*p* < 0.05) and two asterisks (**) highly significant (*p* < 0.01) differences as determined by Student’s *t*-test.

**Figure 4 vaccines-07-00148-f004:**
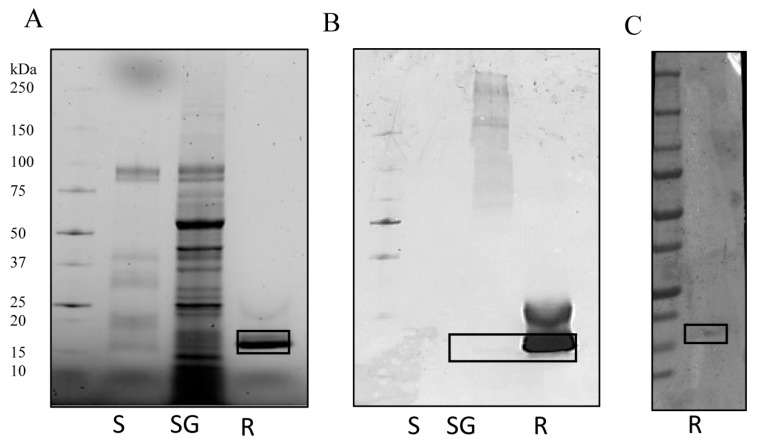
Detection of endogenous IrSPI protein expression. Tick saliva (S) and salivary gland extracts (SG) were obtained from a pool of 17 *I. ricinus* females from the field after a five-day pre-feeding step on rabbits. To each lane, samples of 10 µL of saliva (S), 10 µL of salivary gland extract (SG), and 1.4 µg of recombinant IrSPI (R) were loaded. (**A**) SDS-PAGE electrophoresis and UV visualization. (**B**) Western blot analysis using antiserum from an IrSPI-immunized mouse. (**C**) Western blot analysis using an antiserum from a tick-infested rabbit. Bands corresponding to IrSPI are framed with boxes.

**Figure 5 vaccines-07-00148-f005:**
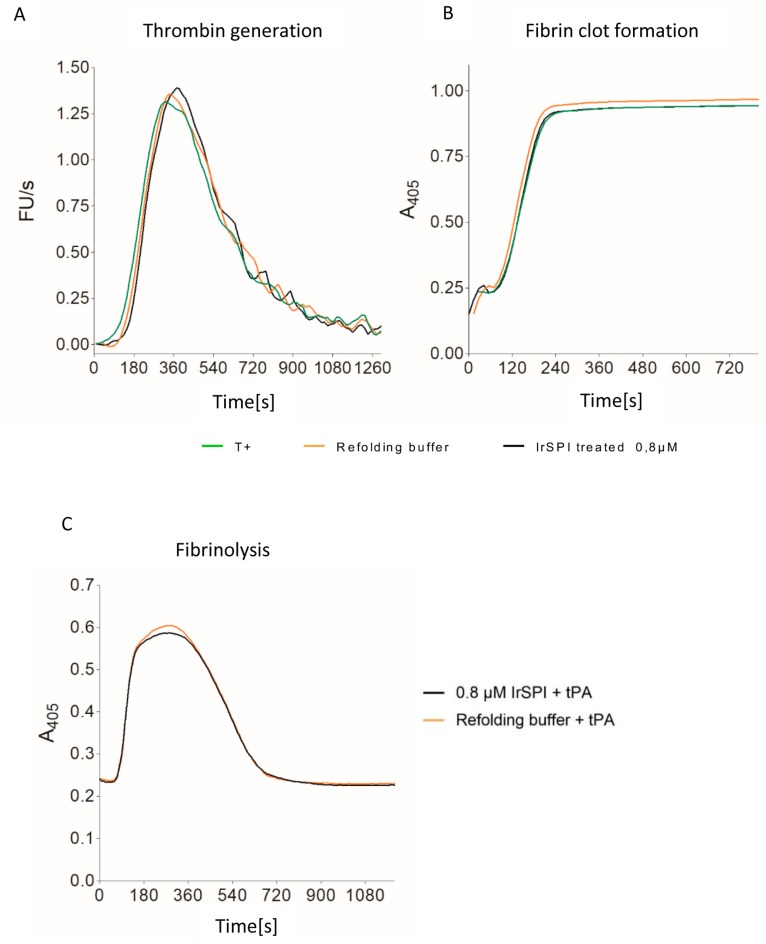
Effect of IrSPI on vertebrate host coagulation. (**A**) Thrombin generation (in fluorescence unit/s) triggered by adding 10 mM CaCl2 to platelet-poor plasma containing tissue factor and phospholipid (PPP-reagent). Curves in black were obtained in the presence of 0.8 µM IrSPI in refolding buffer, curves in orange in refolding buffer only, and curves in green in 0.15 M NaCl. (**B**) Simultaneous recording of the fibrin clot formation detected through the increase of turbidity at 405 nm; legend as in (**A**). (**C**) Fibrin clot formation was triggered as in (**A**,**B**) except that 6 nM tPA was added before clot formation. Half-lysis times (defined as the time to halve maximum turbidity) were not significantly different in the presence (curve in black) or absence (curve in orange) IrSPI. Results presented are mean of two independent experiments.

**Figure 6 vaccines-07-00148-f006:**
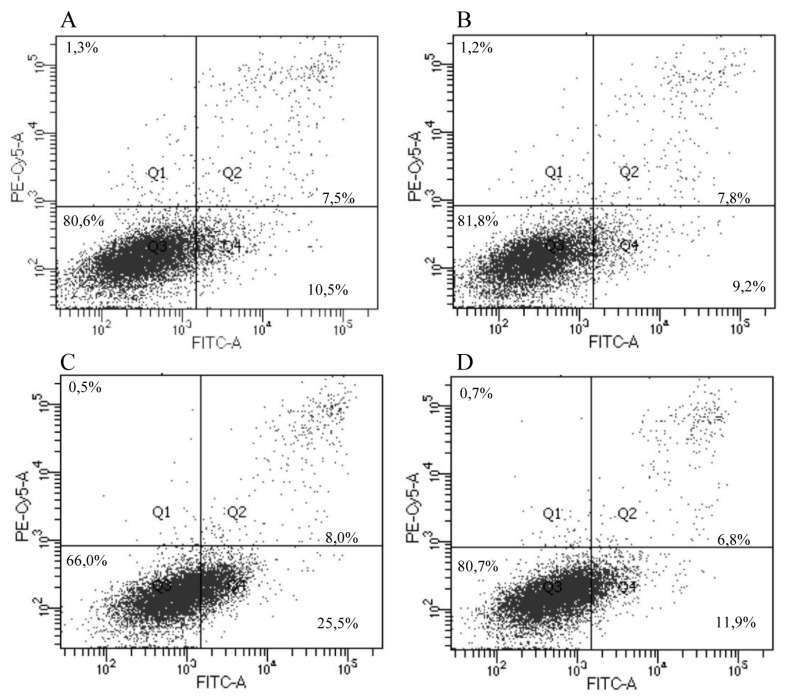
Impact of IrSPI on endothelial cell apoptosis. Viability and apoptosis were monitored by flow cytometry in human skin microvascular endothelial cell line (HSkMEC) endothelial cells after labeling with propidium iodide and Annexin V-FITC, respectively. Cells were incubated for 40 h in the presence of (**A**) culture medium, (**B**) 1 µM of IrSPI without refolding, (**C**) refolding medium, or (**D**) 1 µM of refolded IrSPI. The percentage of apoptotic cells was determined as the number of cells labeled with Annexin V, but not propidium iodide (Q4) out of 10,000 cells analyzed per condition.

**Figure 7 vaccines-07-00148-f007:**
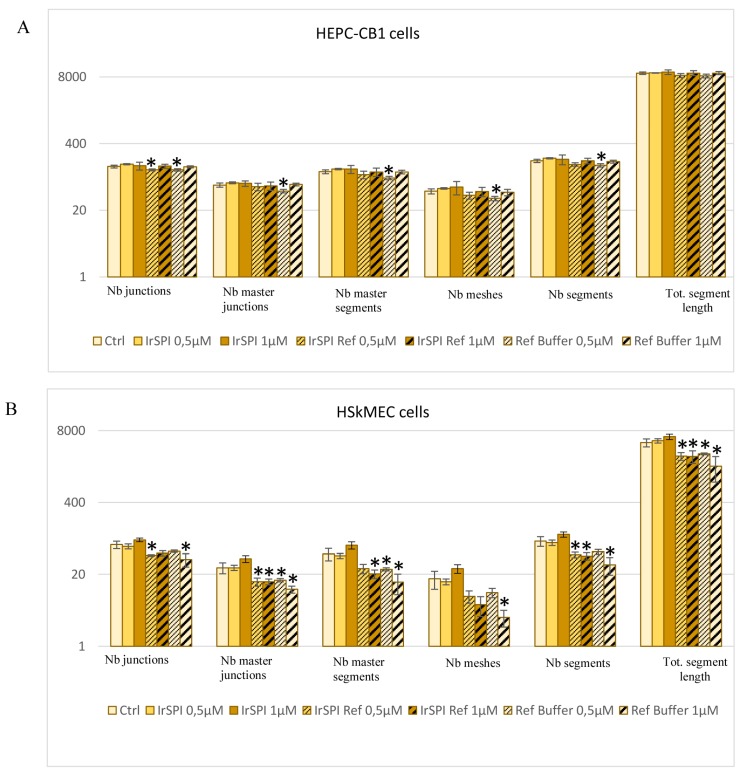
Impact of IrSPI on angiogenesis. (**A**) HEPC-CB1 (human endothelial progenitor cells-cord blood 1) and (**B**) HSkMEC endothelial cells were cultivated with IrSPI prior to (IrSPI), or after a refolding step (IrSPI Ref) for 9.5 h, and the number of junctions, master junctions, segments, master segments, meshes, and total segment lengths were recorded via video-microscopy every 30 min and analyzed with ImageJ software. Results are expressed as means of triplicates. Controls correspond to culture medium (ctrl) and refolding buffer (Ref buffer). Asterisks (*) indicate significant differences as determined using the Student’s *t* test (*p* < 0.05).

**Figure 8 vaccines-07-00148-f008:**
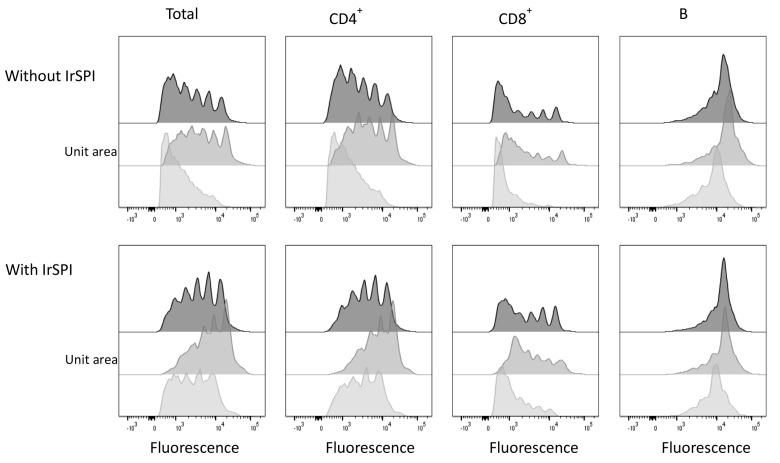
Impact of IrSPI on murine splenocyte proliferation. The impact of IrSPI on the proliferation of T and B cells subsets was evaluated in splenocytes of OF1 mice following mitogenic stimulation by Concanavalin A. Cell proliferation was measured by CFSE dye dilution and flow cytometry. Splenocytes from each of three mice were separately evaluated in triplicate. Results are expressed for each condition as the mean of triplicate wells for each condition. Successive peaks, displaying diminished CFSE labeling, reveal successive cell divisions.

**Table 1 vaccines-07-00148-t001:** Impact of IrSPI on murine splenocyte and macrophage cytokine/chemokine production with or without mitogen stimulation by concanavalin A (ConA).

Conditions	Cytokine/Chemokine	Effect of IrSPI (Percentage)	*p*-Value
		**Splenocytes**	
Without ConA	IP10	−32.8%	2.21 × 10^−3^
	MCP3	−72.3%	8.07 × 10^−4^
	MIP-1β	−26.0%	2.11 × 10^−3^
	RANTES	−15.6%	3.94 × 10^−3^
With ConA	IL-1β	−45.0%	1.15 × 10^−3^
	IL-13	−75.1%	4.47 × 10^−4^
	IL-18	−46.0%	1.63 × 10^−7^
	IL-6	−54.8%	5.80 × 10^−6^
	IL-9	−48.8%	7.06 × 10^−3^
	TNF-α	−46.1%	7.23 × 10^−6^
	IFN-γ	−50.3%	2.28 × 10^−7^
	IP10	−46.5%	4.30 × 10^−5^
	MIP-1β	−17.6%	2.28 × 10^−3^
	RANTES	−37.9%	1.26 × 10^−4^
	Eotaxin	−21.3%	3.89 × 10^−3^
	GM-CSF	−71.9%	8.67 × 10^−8^
	IL-2	+ 49.2%	5.62 × 10^−4^
		**Macrophages**	
With LPS/IFN-γ	IL-5	−9.6%	8.37 × 10^−3^

All results are presented as significant expression changes (*p*-value < 0.01; ANOVA) after a three-day IrSPI incubation. Splenocytes from each of three mice were separately evaluated in triplicate for each condition. Results are expressed as the mean for the three mice after calculating the mean of triplicate wells for each mouse.

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
