# Peer review of "The Immunomodulatory Effect of IrSPI, a Tick Salivary Gland Serine Protease Inhibitor Involved in *Ixodes ricinus* Tick Feeding"

_vaccines, 2019, doi:10.3390/vaccines7040148_

Round 1

Reviewer 1 Report

The manuscript treat of interesting results obtained to IrSPI, a serine protese inhibitor, a great number of results were obtained, for that I congratulate the scientifc group.

However during the reading a doubt comes from to me, why the authors performed the immune modulation experiments only with no refolded IrSPI? Or it was done? Follow some observations about the document.

Line 111 – It is necessary to verify the number of nucleotides, 285 plus (19 x 3 = 57) is 342 not 321.

Line 116 – To verify the number of fig 1 (A or B), to verify the other numbering of fig 1 in the other parts of the text too.

Line 134, In Fig. 1 apparently the dissulfide bonds indicated in the primary structure should be reviewed, other papers indicated different cysteins envolved in each bond.

Follow some references: Protein folding guides disulfide bond formation. Qin M, Wang W, Thirumalai D. Proc Natl Acad Sci U S A. 2015 Sep 8;112(36):11241-6. doi: 10.1073/pnas.1503909112. Epub 2015 Aug 21.

Divergent folding pathways of two homologous proteins, BPTI and tick anticoagulant peptide: compartmentalization of folding intermediates and identification of kinetic traps. Chang JY, Li L. Arch Biochem Biophys. 2005 May 1;437(1):85-95.

Line 145 – The “percentage of identity with IrSPI” is not indicated in the figure.             

Line 148 Serine protease inhibition

Probably the elastase inhibition is under estimated because the quantity of active inhibitor was not measure exactly, because probably only one percentage of the inhibitor was refolded. I recommend to do a dose response curve to show the elastase inhibition activity of IrSPI, and to obtain the Ki value, one possibility is to use the Morrison equation if  the inhibition is slow tight binding type.

Line 154 – Which Kallikrein was not inhibited? Or were Kallikreins?

Line 160 – Clarify in the protocol, if the disulfide isomerase protein and glutathione is extracted from the samples after the refolding procedure, and if not, are the presence of these molecules in the buffer interfere in the inhibition? Discuss about it.

Line 224 – Detection of endogenous IrSPI – I recommend the use of mass spectrometry to confirm the presence of IrSPI in the saliva.

Line 253 – To confirm if is “halve” or “have”.

Line 320 – How is explained the action of the non refolded IrSPI in immune modulation, instead of refolded IrSPI? Was the refolded molecule used in the experiments? In the case of non refolded IrSPI, is there the possibility to exist some IrSPI folded molecules in the sample even previous refolding procedure? Is not clear the possible mechanism of action in the immune modulation.

Author Response

The manuscript treat of interesting results obtained to IrSPI, a serine protese inhibitor, a great number of results were obtained, for that I congratulate the scientifc group.

However during the reading a doubt comes from to me, why the authors performed the immune modulation experiments only with no refolded IrSPI? Or it was done?

And Line 320 – How is explained the action of the non refolded IrSPI in immune modulation, instead of refolded IrSPI? Was the refolded molecule used in the experiments? In the case of non refolded IrSPI, is there the possibility to exist some IrSPI folded molecules in the sample even previous refolding procedure? Is not clear the possible mechanism of action in the immune

This experiment was indeed only performed with the non-refolded recombinant protein for several reasons. First, the use of the reduced form of glutathione generates a toxic compound that causes cell death, second, the elimination of the glutathione after the refolding process may be delicate due to its small size (less than 1 KDa) and, finally, the fact that we did not have a large amount of recombinant protein available. Critically, preliminary experiments using the non-refolded recombinant protein clearly revealed that it inhibited cell proliferation. Thus, the mode of action may be independent of the inhibitory activity of IrSPI on serine proteases and thus of the form (folded or non-folded) of the protein. While, as the reviewer has suggested, we cannot formally exclude the presence of some quantity of folded IrSPI in the non-refolded sample, the results obtained in the protease inhibition assays do not support this hypothesis. Detailed study of the mechanism of action of the non-refolded protein on both splenocytes and macrophages is beyond the scope of the current study, but should be pursued in future studies.

Follow some observations about the document.

Line 111 – It is necessary to verify the number of nucleotides, 285 plus (19 x 3 = 57) is 342 not 321.

Thanks for this remark, this has been corrected in the new version of the manuscript

Line 116 – To verify the number of fig 1 (A or B), to verify the other numbering of fig 1 in the other parts of the text too.

This was changed in the new version in keeping with the new organization of the manuscript

Line 134, In Fig. 1 apparently the dissulfide bonds indicated in the primary structure should be reviewed, other papers indicated different cysteins envolved in each bond. Follow some references: Protein folding guides disulfide bond formation. Qin M, Wang W, Thirumalai D. Proc Natl Acad Sci U S A. 2015 Sep 8;112(36):11241-6. doi: 10.1073/pnas.1503909112. Epub 2015 Aug 21. Divergent folding pathways of two homologous proteins, BPTI and tick anticoagulant peptide: compartmentalization of folding intermediates and identification of kinetic traps. Chang JY, Li L. Arch Biochem Biophys. 2005 May 1;437(1):85-95.

The reviewer is correct and we apologize for the likely incorrect assignment of the disulfide bonds of folded IrSPI proposed in figure 1. Mammalian Kunitz type 1 inhibitors typically have 6 cysteines, namely C5, C14, C30, C38 C51 and C55 (in the original Kunitz numbering system). These cysteines are involved in 3 intramolecular disulfide bonds: C5-C55, C14-C38 and C30-C51. Several Kunitz-type toxin inhibitors (including IrSPI) possess an additional cysteine residue, adjacent to the universally conserved C51. In these toxins, the intra-molecular disulfide bond pattern for the topologically equivalent cysteines is identical. The extra cysteine following C51 is unengaged in intra-molecular bonds (Chang JY, Li L. Arch Biochem Biophys. 2005; 437:85-95; Chen Z1, Cao Z, Li W, Wu Y, Toxicon. 2013; 72:5-10).

We thank the reviewer for her/his relevant comment and have corrected Figure 1 accordingly.

Line 145 – The “percentage of identity with IrSPI” is not indicated in the figure.   

It was indeed an oversight that was corrected in the new version.       

Line 148 Serine protease inhibition

Probably the elastase inhibition is under estimated because the quantity of active inhibitor was not measure exactly, because probably only one percentage of the inhibitor was refolded. I recommend to do a dose response curve to show the elastase inhibition activity of IrSPI, and to obtain the Ki value, one possibility is to use the Morrison equation if the inhibition is slow tight binding type.

We agree with reviewer 1 that inhibition of elastase is probably underestimated. Residual activity of elastase was measured following 15 min incubation with IrSPI at 37°C. Thus equilibrium must have been reached unless the association rate constant is less than 10² M-1 s-1 which would be unexpected for a Kunitz-type 1 inhibitor. Assuming that the Ki value of IrSPI for elastase is in the nM range, our result would reflect titration of “active” IrSPI by elastase. There the 23.9% inhibition observed would indicate that the active (refolded) IrSPI concentration was only 5 nM, partially inhibiting the 21.6 nM added elastase (i.e. only 0.5% of the 1 µM total IrSPI was active). Alternatively, assuming that the 1 µM refolded IrSPI was fully "active" our result would indicate that the Ki value of IrSPI for elastase is quite high (about 3.3 µM). As suggested by reviewer, a dose response curve would permit estimation of the Ki value, whereas slow-tight binding kinetic would in addition provide an estimate of the association rate constant. Unfortunately, performing such estimation would require IrSPI amount currently not available (dose response curve) or prior knowledge of the true “active” concentration of IrSPI (slow-tight binding) currently not available either.”

Line 154 – Which Kallikrein was not inhibited? Or were Kallikreins?

As mentioned in the M &M section, the Kallikrein tested here was from porcine pancreas shipped from Sigma-Aldrich. In the revised version of the manuscript we hope that this point will be clearer.

Line 160 – Clarify in the protocol, if the disulfide isomerase protein and glutathione is extracted from the samples after the refolding procedure, and if not, are the presence of these molecules in the buffer interfere in the inhibition? Discuss about it.

Neither the disulfide isomerase protein nor glutathione were extracted from the samples after the refolding procedure. To avoid any confusion, this point was highlighted in the new version both in the M & M and in the legend of FIG 2. Regarding the experiment with the refolded IrSPI, and as mentioned in the M & M section: “refolded recombinant IrSPI in the refolding buffer, this last buffer being used as a negative control for this latter condition”. Thus, we corrected for any impact that the buffer constituents may have had on inhibition of serine proteases activity by using the buffer as a blank.

Line 224 – Detection of endogenous IrSPI – I recommend the use of mass spectrometry to confirm the presence of IrSPI in the saliva.

We agree with the reviewer that using mass spectrometry would have been a technique of choice to validate the presence of IrSPI in saliva and should be carried out in the future. Unfortunately, we did not have enough biological material to carry out this type of experiment, as all the available saliva was used for the western blot. As mentioned in the MS, under our conditions each tick provides only around 3 µl of saliva over a 6 hour period and after a pre-feeding step.  

Line 253 – To confirm if is “halve” or “have”.

We confirm that “halve” is the right word here.

Reviewer 2 Report

The MS deals with the new serine protease inhibitor IrSPI. Molecular and biochemical characterization of recombinant protein and its functional analysis were performed. Even though IrSPI effect on blood coagulation, fibrinolysis, endothelial cell angiogenesis or apoptosis has not been proven, the protein exerted immunomodulatory and indirect anti-inflammatory activity. The MS represents contribution to the array of immunomodulatory components of tick saliva that were more or less characterized. This is a valuable contribution, because just recognition of the majority of immunomodulatory and anti-inflammatory molecules enables to design a cocktail vaccine having the chance to protect the host from transmission of tick-borne pathogens.

The MS is very well written, Material and methods are given in detail, and results are well documented followed with comprehensive discussion.

Comments and questions:

23.9% inhibition of elastase is not very high in comparison with other Kunitz serine protease inhibitors from ticks (Cao et al 2013, Soares et al. 2016).

Expression of mRNA for IrSPI in nymphs was observed in the very end of the feeding process (the 5th day after the feeding commencement). Could authors discuss the significance of this immunomodulatory protein for the tick feeding process?

Why only non-refolded IrSPI was used in splenocyte proliferation assay? Using the refolded protein with some serine protease inhibitory activity would have better chance to exert an immunomodulatory effect.

For activation of splenocytes, only T-mitogen ConA was used. IrSPI effect on B lymphocytes could be only indirect, owing to changes in production/activity of T lymphocyte cytokines influencing B cell proliferation (IL-4). But in any case B lymphocytes should be activated for example by LPS.

IrSPI inhibited both Th1 and Th2 cytokines, no upregulation of any Th2 cytokine was observed (some Th2 cytokines were not affected). The only upregulated cytokine was IL-2, which is typical Th1 cytokine. Under these conditions, it is difficult to claim that IrSPI skews the Th1/Th2 balance to Th2 response. What is obvious is that the IrSPI effect is anti-inflammatory, facilitating tick feeding and pathogen transmission.

Minor comments:

Page 5, line 124  (Fig 1C) should be changed for (Fig1B)

Page 17, line 329  GM-CSF is not a chemokine

Page 17, line 331  IL-1b is not T helper cytokine

Page 25, line 534  …decreased IL-2 expression in tick saliva… it should be due to the effect of tick saliva?

Page 31, line 681  Three pathogen-free I. ricinus females were nano-injected with 250 µl (1 × 107 CFU) of bacterial suspension. It should be 250 nl.

Author Response

The MS deals with the new serine protease inhibitor IrSPI. Molecular and biochemical characterization of recombinant protein and its functional analysis were performed. Even though IrSPI effect on blood coagulation, fibrinolysis, endothelial cell angiogenesis or apoptosis has not been proven, the protein exerted immunomodulatory and indirect anti-inflammatory activity. The MS represents contribution to the array of immunomodulatory components of tick saliva that were more or less characterized. This is a valuable contribution, because just recognition of the majority of immunomodulatory and anti-inflammatory molecules enables to design a cocktail vaccine having the chance to protect the host from transmission of tick-borne pathogens.

The MS is very well written, Material and methods are given in detail, and results are well documented followed with comprehensive discussion.

Comments and questions:

23.9% inhibition of elastase is not very high in comparison with other Kunitz serine protease inhibitors from ticks (Cao et al 2013, Soares et al. 2016).

We agree and have thus added the following sentences in the MS in the discussion section: “We showed that IrSPI, harboring Ala as P1 residue, does indeed inhibit elastase although in a smaller proportion than what is reported in the literature for other tick kunitz protease inhibitors 33, 34.”

However, it should be noticed that such an inhibition may be underestimate here as residual activity of elastase was measured following 15 min incubation with IrSPI at 37°C. Thus, equilibrium must have been reached unless the association rate constant is less than 10² M-1 s-1 which would be quite unusual for a Kunitz-type 1 inhibitor. Assuming that Ki value of IrSPI for elastase is in the nM range, our result would reflect titration of “active” IrSPI by elastase. There the 23.9% inhibition observed would indicate that active (refolded) IrSPI concentration was only 5 nM, partially inhibiting the 21.6 nM added elastase (i.e. only 0.5% of the 1 µM total IrSPI was active). Alternatively, assuming that the 1 µM refolded IrSPI was fully "active" our result would indicate that the Ki value of IrSPI for elastase is quite high (about 3.3 µM).

Expression of mRNA for IrSPI in nymphs was observed in the very end of the feeding process (the 5th day after the feeding commencement). Could authors discuss the significance of this immunomodulatory protein for the tick feeding process?

Thanks for this suggestion. The following sentence has thus been added in the discussion of the new version of the MS : “The immunomodulatory activity of IrSPI that we observed—that is, reduction in T cell proliferation and more particularly in proliferation of the CD4+ T subset—is compatible with late expression of IrSPI, as the T-cell response is expected to be initiated following the early inflammatory response. “

Why only non-refolded IrSPI was used in splenocyte proliferation assay? Using the refolded protein with some serine protease inhibitory activity would have better chance to exert an immunomodulatory effect.

This experiment was only performed with the non-refolded recombinant protein for several reasons. First, the use of the reduced form of glutathione generates a toxic compound that causes cell death, second, the elimination of the glutathione after the refolding process may be delicate due to its small size (less than 1 KDa) and, finally, the fact that we did not have a large amount of recombinant protein available. Critically, preliminary experiments using the non-refolded recombinant protein clearly revealed that it inhibited cell proliferation. Thus, the mode of action may be independent of the inhibitory activity of IrSPI on serine proteases and thus of the form (folded or non-folded) of the protein. Nevertheless, we agree that the immunomodulatory power of the refolded form of IrSPI should be pursued in future studies.

For activation of splenocytes, only T-mitogen ConA was used. IrSPI effect on B lymphocytes could be only indirect, owing to changes in production/activity of T lymphocyte cytokines influencing B cell proliferation (IL-4). But in any case B lymphocytes should be activated for example by LPS.

We agree. Only ConA was used as we were primarily interested in the T-cell response. Further studies will be required to address the impact of IrSPI on the B-cell response.

IrSPI inhibited both Th1 and Th2 cytokines, no upregulation of any Th2 cytokine was observed (some Th2 cytokines were not affected). The only upregulated cytokine was IL-2, which is typical Th1 cytokine. Under these conditions, it is difficult to claim that IrSPI skews the Th1/Th2 balance to Th2 response. What is obvious is that the IrSPI effect is anti-inflammatory, facilitating tick feeding and pathogen transmission.

We agree, and have modified the manuscript to that effect.

Minor comments:

Page 5, line 124  (Fig 1C) should be changed for (Fig1B)

We apologize for this mistake that has been corrected in the new version

Page 17, line 329  GM-CSF is not a chemokine

We agree and it has been corrected in the new version

Page 17, line 331  IL-1b is not T helper cytokine

We apologize for this mistake that has been corrected in the new version

Page 25, line 534  …decreased IL-2 expression in tick saliva… it should be due to the effect of tick saliva?

We agree and it has been corrected in the revised version of the MS “ This result was somewhat unexpected in the light of published data reporting decreased IL-2 expression by tick saliva”

Page 31, line 681  Three pathogen-free I. ricinus females were nano-injected with 250 µl (1 × 107 CFU) of bacterial suspension. It should be 250 nl.

It has been corrected in the revised version of the MS

Reviewer 3 Report

This article describes the molecular, biochemical and functional characterization of a recently discovered salivary serine protease inhibitor of the Ixodes ricinus tick (IrSPI).

The authors demonstrate that the IrSPI mRNA increases in salivary glands during blood feeding and attempt to demonstrate the presence of the IrSPI protein in the tick saliva.

After that, working with a recombinant form of the IrSPI -produced in insect cells and subjected to refolding treatment-, the authors present convincing experimental results demonstrating that the refolded recombinant IrSPI inhibits elastase and modulates the host immune response without apparently impact on host blood coagulation, apoptosis and angiogenesis.

The  authors conclude that IrSPI is a vital component of the tick saliva inoculated into the host during feeding helping ticks to overcome the host immune defences and successfully feed and transmit pathogens. This converts IrSPI an interesting target for vaccines aimed at the control of tick infestations and tick-borne pathogens.

In summary, this is a good piece of work, but I have some concern regarding demonstration of IrSPI protein in tick saliva. Results presented in fig 4 are not conclusive since the anti-recombinant serum does not recognize native the IrSPI protein in tick saliva -as the own authors acknowledge- and, conversely, the serum against the native IrSPI protein -from an infested rabbit- does not seem to recognize the recombinant protein; actually, the spot boxed by the authors can hardly be considered a real band.

Western blot analysis of saliva samples taken at feeding times longer than 5 days (when mRNA peaks) could perhaps reveal the presence of the protein. Additionally, the 5 days saliva can be subjected to proteomic analysis for IrSPI identification.

Consequently, if no additional convincing evidence is provided, the statements regarding this issue in the discussion and conclusions sections should be moderated: lines 362-363, lines 422-423, lines 551-552.

Minor issues:

Fig. 1. The A, B and C panels seem to be wrongly cited in the text in pag. 5. The identity percentages with IrSPI in panel C are missing. Check for some minor grammar and spell errors. For example: line 423: …this DEMONSTRATES…; line 567: …should ideally BE evaluated…

Author Response

This article describes the molecular, biochemical and functional characterization of a recently discovered salivary serine protease inhibitor of the Ixodes ricinus tick (IrSPI).

The authors demonstrate that the IrSPI mRNA increases in salivary glands during blood feeding and attempt to demonstrate the presence of the IrSPI protein in the tick saliva.

After that, working with a recombinant form of the IrSPI -produced in insect cells and subjected to refolding treatment-, the authors present convincing experimental results demonstrating that the refolded recombinant IrSPI inhibits elastase and modulates the host immune response without apparently impact on host blood coagulation, apoptosis and angiogenesis.

The  authors conclude that IrSPI is a vital component of the tick saliva inoculated into the host during feeding helping ticks to overcome the host immune defences and successfully feed and transmit pathogens. This converts IrSPI an interesting target for vaccines aimed at the control of tick infestations and tick-borne pathogens.

In summary, this is a good piece of work, but I have some concern regarding demonstration of IrSPI protein in tick saliva. Results presented in fig 4 are not conclusive since the anti-recombinant serum does not recognize native the IrSPI protein in tick saliva -as the own authors acknowledge- and, conversely, the serum against the native IrSPI protein -from an infested rabbit- does not seem to recognize the recombinant protein; actually, the spot boxed by the authors can hardly be considered a real band.

We apologize for the quality of the photo of the western blot shown in figure 4 that we were unable to redo due to lack of biological material. Nevertheless, we assure the reviewer, that a band was well detected by the rabbit serum on this western blot.

Western blot analysis of saliva samples taken at feeding times longer than 5 days (when mRNA peaks) could perhaps reveal the presence of the protein. Additionally, the 5 days saliva can be subjected to proteomic analysis for IrSPI identification.

We agree with the reviewer that using mass spectrometry would have been a technique of choice to validate the presence of IrSPI in saliva and should be carried out in the future. Unfortunately, we did not have enough biological material to carry out this type of experiment as all available saliva was used for the western blot.

Consequently, if no additional convincing evidence is provided, the statements regarding this issue in the discussion and conclusions sections should be moderated: lines 362-363, lines 422-423, lines 551-552.

In response to the reviewer's comment, the following sentences have been added in the new version of the MS in the discussion section: “IrSPI is a Kunitz SPI that is induced during blood feeding and appears to be present in tick saliva, even if it will have to be validated by proteomic studies.” And “native IrSPI protein may be present in tick saliva and injected into the host during feeding.”, as well as in the conclusion: “IrSPI is produced by the most common tick vector in Europe, and seems to be a vital component of the tick salivary cocktail”

Minor issues:

Fig. 1. The A, B and C panels seem to be wrongly cited in the text in pag. 5.

We apologize for this mistake that has been corrected in the new version

The identity percentages with IrSPI in panel C are missing.

This has been added in the new version of figure 1

Check for some minor grammar and spell errors.

The manuscript has been carefully reread and we hope that no errors remain.

For example: line 423: …this DEMONSTRATES…; line 567: …should ideally BE evaluated…

This has been corrected